# Dysregulation of sonic hedgehog signaling causes hearing loss in ciliopathy mouse models

Kyeong-Hye Moon[1,2], Ji-Hyun Ma[1], Hyehyun Min[1], Heiyeun Koo[1,2], HongKyung Kim[1], Hyuk Wan Ko[3]*, Jinwoong Bok[1,2,4]*

[1]Department of Anatomy, Yonsei University College of Medicine, Seoul, Republic of Korea; [2]BK21 PLUS project for Medical Science, Yonsei University College of Medicine, Seoul, Republic of Korea; [3]Department of Biochemistry, College of Life Science and Biotechnology, Yonsei University, Seoul, Republic of Korea; [4]Department of Otorhinolaryngology, Yonsei University College of Medicine, Seoul, Republic of Korea

**Abstract** Defective primary cilia cause a range of diseases known as ciliopathies, including hearing loss. The etiology of hearing loss in ciliopathies, however, remains unclear. We analyzed cochleae from three ciliopathy mouse models exhibiting different ciliogenesis defects: *Intraflagellar transport 88* (*Ift88*), *Tbc1d32* (a.k.a. *bromi*), and *Cilk1* (a.k.a. *Ick*) mutants. These mutants showed multiple developmental defects including shortened cochlear duct and abnormal apical patterning of the organ of Corti. Although ciliogenic defects in cochlear hair cells such as misalignment of the kinocilium are often associated with the planar cell polarity pathway, our results showed that inner ear defects in these mutants are primarily due to loss of sonic hedgehog signaling. Furthermore, an inner ear-specific deletion of *Cilk1* elicits low-frequency hearing loss attributable to cellular changes in apical cochlear identity that is dedicated to low-frequency sound detection. This type of hearing loss may account for hearing deficits in some patients with ciliopathies.

*For correspondence:
kohw@yonsei.ac.kr (HWK);
bokj@yuhs.ac (JB)

Competing interests: The authors declare that no competing interests exist.

## Introduction

The primary cilium is a microtubule-based organelle protruding from the apical surface of nearly all animal cells (*Goetz and Anderson, 2010*). This microtubular projection is anchored by the basal body and enveloped by the plasma membrane. The formation and maintenance of primary cilia are regulated by a dynamic transport system known as intraflagellar transport (IFT), which is the bidirectional cargo transport system gated by ciliary appendages (*Goetz and Anderson, 2010*). The ciliary cargos include proteins that make up the cilium, as well as various signaling receptors and effectors. Such enrichment of various signaling components within a slender cellular apparatus may be a good strategy to improve sensitivity to receiving signals, making the primary cilium an efficient receiving center (*Goetz and Anderson, 2010*; *Pala et al., 2017*).

Primary cilia are known to play crucial roles in animal development and homeostasis by mediating various signaling pathways including sonic hedgehog (SHH), WNT, platelet-derived growth factor (PDGF), TGF-β, Notch, and mechanosensation (*Christensen et al., 2017*; *Nishimura et al., 2019*). The planar cell polarity (PCP) pathway, which is associated with non-canonical WNT signaling, has also been implicated with primary cilia (*Wallingford, 2012*). Thus, gene mutations affecting ciliary formation and function elicit a broad spectrum of genetic disorders known as ciliopathies (*Nishimura et al., 2019*; *Reiter and Leroux, 2017*). Hearing loss is one of the sensorineural defects commonly observed in ciliopathies (*Esposito et al., 2017*; *Lindsey et al., 2017*; *Reiter and Leroux,*

*2017*; *Ross et al., 2005*), yet the etiology of hearing loss in relation to ciliary defects remains unclear.

The organ of Corti is the peripheral auditory sensory organ of the inner ear, comprised of mechanosensory hair cells (HCs) and various supporting cells (SCs). Primary cilia are present in both HCs and SCs, with those in HCs known as kinocilia. The role of kinocilia has been highlighted by their polarized migration to the abneural (lateral) side of the HC, which guides the subsequent orientation of actin filament-rich stereociliary (hair) bundles. Thus, a hair bundle, arranged in a three-row staircase pattern, is positioned such that the tallest row is closest to the kinocilium (Figure 2A). Notably, the hair bundle of outer hair cells (OHCs) is V-shaped, with its vertex located directly adjacent to the kinocilia at the abneural side (Figure 2A). Mutants lacking kinocilia due to mutations in genes encoding the IFT complex such as *Ift88*, *Ift20*, and *Kif3a* exhibit abnormal hair bundle orientation and morphology as well as random localization of the basal body, which forms the basis for the kinocilium (*Jones et al., 2008*; *May-Simera et al., 2015*; *Sipe and Lu, 2011*). The kinocilium-directed uniform orientation of hair bundles is considered essential for efficient responses to sound vibrations and normal hearing. However, recent studies have indicated that defective hair bundle polarity is not always associated with hearing loss, as is observed in *Ttc8*$^{-/-}$ (a.k.a. *Bbs8*$^{-/-}$) and *Alms1*$^{-/-}$ mutant mice, in which hearing is normal until several months after birth despite evident neonatal polarity defects (*Collin et al., 2005*; *Jagger et al., 2011*; *May-Simera et al., 2015*). These results suggest that polarity defects may not account for all instances of hearing loss observed in ciliopathies.

Other than polarity defects, mutants lacking primary cilia exhibit severe shortening of cochlear ducts, which may lead to hearing loss due to cochlear hypoplasia or Mondini disorders (*Jones et al., 2008*; *May-Simera et al., 2015*; *Montcouquiol and Kelley, 2020*; *Sipe and Lu, 2011*). Considering the role of primary cilia in PCP signaling, which provides the directionality for convergent extension (*Rida and Chen, 2009*; *Wallingford, 2010*), cochlear shortening in ciliary mutants could result from abnormal PCP signaling. However, research has shown that *Tmem67* and *Bbs8* (a.k.a. *Ttc8*) mutant ciliopathy models or inner ear-specific *Vangl2* mutants showed normal cochlear lengths despite exhibiting severe polarity defects (*Abdelhamed et al., 2015*; *Copley et al., 2013*; *May-Simera et al., 2015*). These data suggest that PCP signaling may be uncoupled from cochlear extension and that there may be an additional role of primary cilia in promoting cochlear extension.

In this study, to better understand the role of primary cilia in the cochlea, we analyzed cochlear phenotypes in three different categories of ciliopathy mouse models, including no cilia (*Ift88* mutants), morphologically defective cilia (*Tbc1d32*$^{bromi}$ mutants), and abnormally elongated cilia (*Cilk1* mutants) (*Jones et al., 2008*; *Ko et al., 2010*; *Moon et al., 2014*). Comparative analysis of our results and previous studies suggest that there are two categories of cochlear phenotypes in ciliopathy models: one associated with impaired SHH signaling and the other without SHH-associated defects. Our work suggests that dysregulation of SHH signaling is an important consideration for the etiology of hearing loss in ciliopathies.

## Results

### Primary cilia are required for SHH signal transduction in the inner ear primordium

To investigate the role of primary cilia in inner ear development, we first examined whether primary cilia are present in the otocyst, or inner ear primordium, at embryonic day (E) 10.5 (*Figure 1*). Primary cilia were visualized by immunostaining with antibodies against ARL13B and γ-tubulin, which are localized to the ciliary membrane and basal body, respectively (*Cantagrel et al., 2008*). Primary cilia were observed in otic epithelium projecting toward the lumen of the otocyst (*Figure 1B*, Ba, C) and in the periotic mesenchyme (*Figure 1Bb*). The abundance of primary cilia appeared to be proportional to the thickness of the otic epithelium, higher in the ventral and medial regions than in the dorsolateral region (*Figure 1B,C*; *Figure 1—figure supplement 1*). SHH target genes including *Ptch1* and *Gli1* are most strongly expressed in the ventral and medial regions, closer to the SHH sources such as the floor plate and notochord (*Figure 1G'*), than in the dorsal and lateral regions (*Figure 1G–I*). Interestingly, the abundance of ciliated cells in each otic region is generally correlated with the expression levels of SHH target genes (*Figure 1—figure supplement 1*).

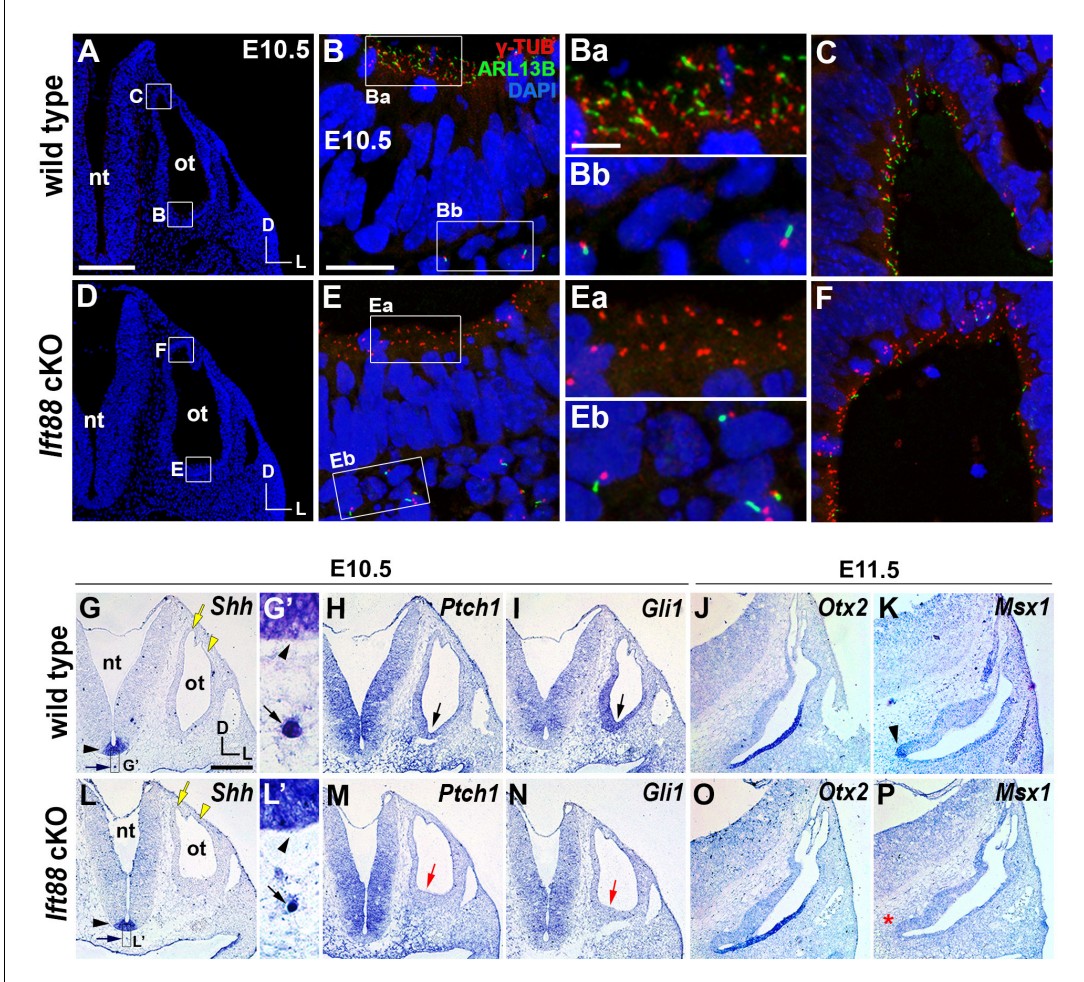

**Figure 1.** Sonic hedgehog signaling is affected in *Ift88* cKO otocysts. (A–C) In E10.5 wild-type embryos, primary cilia visualized by ARL13B and γ-tubulin immunostaining were observed in the otic epithelium (B, Ba, C) and periotic mesenchyme (Bb). (D–F) In *Ift88* cKO embryos, primary cilia were disappeared in the otic epithelium (E, Ea, F) but not in periotic mesenchyme (Eb). (G–I, L–N) *Shh* is expressed in the floor plate (G, G', L, L'; black arrowheads) and notochord (G, G', L, L'; black arrows) in both E10.5 wild-type and *Ift88* cKO embryos. Yellow arrows and arrowheads indicate the endolymphatic duct and vertical pouch, respectively. The sonic hedgehog (SHH) target genes, *Ptch1* and *Gli1,* are expressed in a graded pattern, stronger in the ventral and medial regions and weaker in the dorsolateral region of the otocyst in wild-type embryos (H, I), but dramatically decreased in *Ift88* cKO embryos (M, N; red arrows). (J, K) In E11.5 wild-type otocysts, *Otx2* is expressed in the lateral side of the developing cochlea and *Msx1* is expressed in the apical tip of the cochlea (K; arrowhead). (O, P) In *Ift88* cKO otocysts, while *Otx2* expression appears normal, *Msx1* expression is completely downregulated (P; red asterisk). nt, neural tube; ot, otocyst. In all images, dorsal is up and lateral is right. Scale bar in A, 200 µm, also applies to D; scale bar in B, 20 µm, also applies to C, E, F; scale bar in Ba, 3 µm, also applies to Bb, Ea, Eb; scale bar in G, 200 µm, also applies to H–P. The online version of this article includes the following source data and figure supplement(s) for figure 1:

**Figure supplement 1.** Quantification of cilia number and sonic hedgehog (SHH) target gene expression in the inner ear.

**Figure supplement 1—source data 1.** Raw data for cilia number and in situ hybridization signal intensity shown in *Figure 1—figure supplement 1*.

To examine the role of primary cilia in the otocyst, we ablated primary cilia in the otic epithelium by crossing *Ift88^{fl/fl}* mice with *Pax2-Cre* mice. IFT88 is an IFT-B component that is essential for anterograde transport along the cilia (*Pazour et al., 2000*), and *Pax2-Cre* mice induce Cre-mediated recombination in the otic placode as early as six to seven somite stages (*Ohyama and Groves, 2004*). In E10.5 *Pax2-Cre; Ift88^{fl/fl}* (*Ift88* conditional knockout [cKO]) embryos, primary cilia were specifically ablated in the otic epithelium (*Figure 1Ea*), but not in the periotic mesenchyme (*Figure 1Eb*), compared to controls (*Figure 1A–C*). Although *Shh* expression in the ventral midline structures, floor plate and notochord, was not downregulated in *Ift88* cKO embryos (*Figure 1G,L*; black arrowhead and arrow), SHH target genes including *Ptch1* and *Gli1* were significantly

downregulated in the otic epithelium but not in the periotic mesenchyme (*Figure 1H,I,M,N*; *Figure 1—figure supplement 1*). The dorsal otocyst showed relatively normal morphology with the endolymphatic duct and vertical pouch (*Figure 1G,L*; yellow arrow and arrowhead); however, the ventral tip of the cochlear primordium was not as prominent as that in wild-type embryos (*Figure 1H,I,M,N*; black and red arrows). Consistent with these results, expression of *Msx1* at the tip of apical cochlea, which requires strong SHH signaling (*Figure 1K*, arrowhead, *Son et al., 2015*), was not detected in *Ift88* cKO embryos at E11.5 (*Figure 1P*; red asterisk). In contrast, expression of *Otx2*, which requires a relatively lower level of SHH signaling than *Msx1* (*Bok et al., 2007*; *Morsli et al., 1999*; *Riccomagno et al., 2002*), was comparable between *Ift88* cKO and wild-type embryos (*Figure 1J,O*). These results suggest that primary cilia are required for normal SHH signal transduction in the inner ear primordium, particularly the strong SHH activity required for apical cochlear patterning.

## Cochlear phenotypes in three different types of ciliary mutants

Next, we examined the role of primary cilia in cochlear development and HC differentiation by analyzing the cochlear phenotypes of three different types of ciliary mutants, whose ciliogenesis phenotypes have been characterized in other tissues: (1) *Pax2-Cre; Ift88^{fl/fl}* (*Ift88* cKO) mutant lacking primary cilia (*Haycraft et al., 2007*; *Jones et al., 2008*), (2) *Tbc1d32^{bromi}* mutant, generated by ENU-mutagenesis and exhibiting defective ciliary morphology such as swollen or bulbous cilia (*Ko et al., 2010*), and (3) *Cilk1* (also called *Ick*) knockout mutant exhibiting abnormal ciliary length (*Moon et al., 2014*). Cochleae were dissected at E18.5 and stained with phalloidin and anti-ARL13B antibody to visualize stereociliary bundles and primary cilia, respectively (*Figure 2*).

In wild-type mice, the cochlear length reached one and a three-quarter turns by E18.5 (*Figure 2B*), and the organ of Corti showed a progression of HC differentiation from base to apex, with morphologically more organized mature HCs at the base (*Figure 2C*), compared to less organized immature HCs toward the apex (*Figure 2D,E*). Kinocilia were present at the vertex of V-shaped stereociliary bundles in mature HCs at the base (*Figure 2Ca*, Cb; yellow arrows) and in the center of immature HCs at the apex (*Figure 2Ea*; yellow arrows). Primary cilia were found in SCs, such as Deiters' cells (*Figure 2Ca*; white arrows) and pillar cells (*Figure 2Cb*; white arrowheads).

In *Ift88* cKO mutants, kinocilia in HCs and primary cilia in SCs were mostly absent (*Figure 2F–J*). As previously reported (*Jones et al., 2008*), *Ift88* cKO mutants exhibited shortened cochlear ducts with multiple extra rows of HCs at the apex and severe hair bundle polarity defects (*Figure 2F–I,U*). Total HC number was significantly reduced proportional to the decreased cochlear length (*Figure 2U,V*). We also observed that HCs were prematurely differentiated based on the hair bundle morphology of the apical HCs, compared to wild types (*Figure 2E,I*). Interestingly, ectopic vestibule-like HCs were observed in the non-sensory region of Kölliker's organ in *Ift88* cKO cochlea (*Figure 2J*; n = 4/4).

In *Tbc1d32^{bromi}* mutants, cochlear length was also shortened (*Figure 2K,U*), with multiple extra rows of HCs at the apex (*Figure 2M,N*) and with a significant decrease in total HC number proportional to the shortened cochlear length (*Figure 2U,V*). Kinocilia were significantly elongated, compared with wild-type cochlea (*Figure 2T*), and ciliary tips were often swollen (*Figure 2—figure supplement 1A–F*, red arrow). Despite the presence of kinocilia in HCs, ARL13B-positive primary cilia were not detected in most SCs including Deiters' cells (*Figure 2La*; asterisks) and pillar cells (*Figure 2Lb*; asterisks). The presence of SCs was confirmed by SOX2 immunostaining (*Figure 2—figure supplement 1G–L*). Similar to *Ift88* cKO, *Tbc1d32^{bromi}* mutants exhibited premature HC differentiation (*Figure 2N*), compared to wild type (*Figure 2E*). Similar to *Ift88* cKO, *Tbc1d32^{bromi}* mutants also showed ectopic vestibule-like HCs in the non-sensory region of the Kölliker's organ (*Figure 2O*; n = 3/4) and adjacent to inner hair cells (IHCs) (*Figure 2M*; yellow arrowheads).

In *Cilk1* KO mutants, both kinocilia of HCs and primary cilia of SCs were significantly elongated compared with those in wild-type cochlea (*Figure 2Q–S,T*). Some of the SCs from *Cilk1* KO mutants lacked primary cilia (*Figure 2Qa*; asterisk). Cochlear length was significantly shortened, but longer than those in *Ift88* cKO and *Tbc1d32^{bromi}* mutants (*Figure 2P,U*), and the total number of HCs was also higher (*Figure 2V*). HCs appeared slightly more mature than wild type based on a more organized pattern and stereociliary bundles in the apex (*Figure 2S*).

Due to the shortened cochlear length of ciliary mutants, the absolute distances of 90% of the length from the basal end of *Tbc1d32^{bromi}*, *Ift88*, and *Cilk1* mutant cochlea correspond to 52%, 58%,

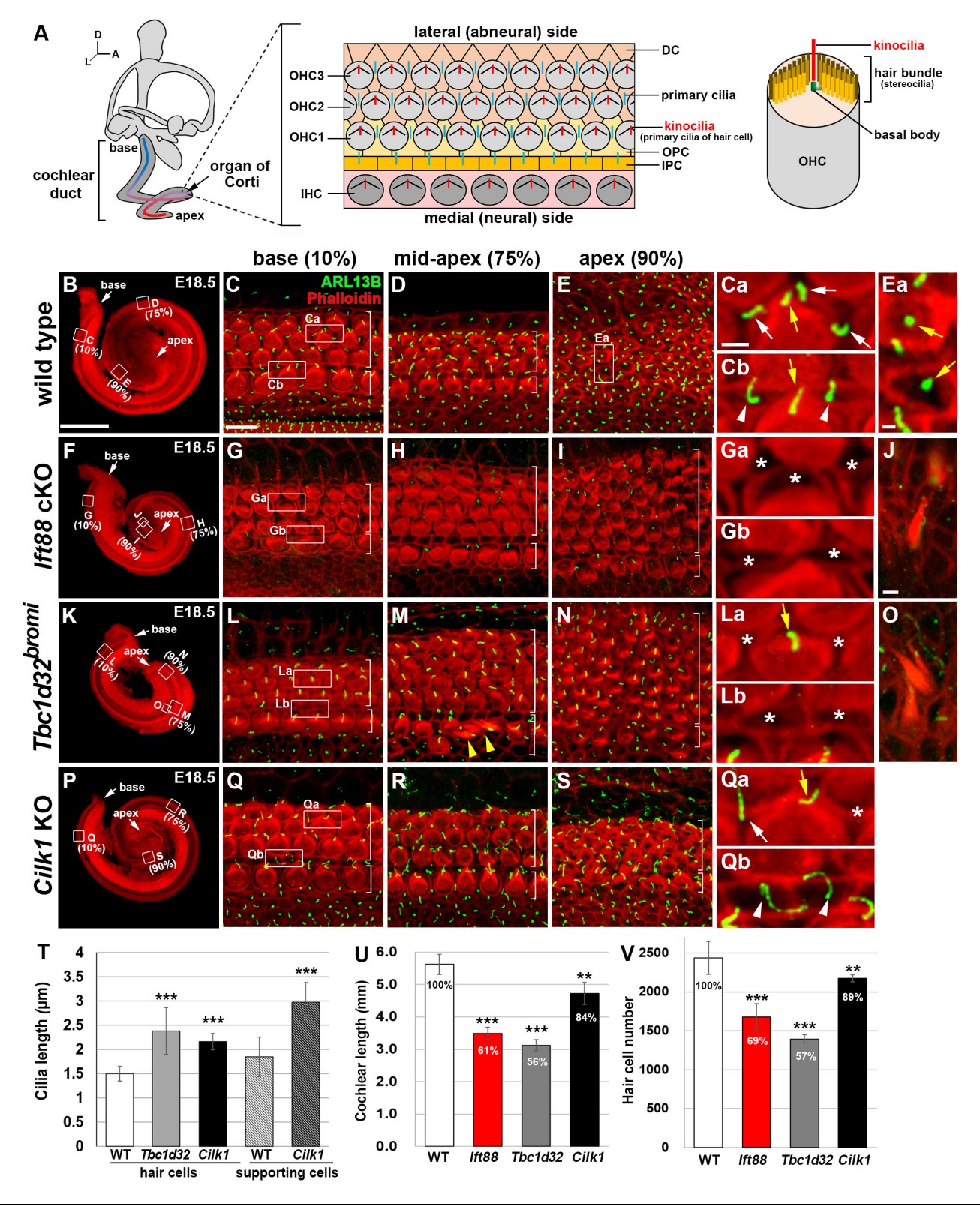

**Figure 2.** Cochlear phenotypes of three ciliary mutants at E18.5. (A) Schematic diagram of E18.5 organ of Corti. OHC, outer hair cell; IHC, inner hair cell; DC, Deiter's cell; OPC, outer pillar cell; IPC, inner pillar cell. (B–E) In E18.5 wild-type cochlea, stereociliary bundles and primary cilia were visualized by staining with phalloidin (red fluorescence) and immunostaining with anti-ARL13B antibody (green fluorescence), respectively. Images of organ of Corti from 10% (base), 75% (mid-apex), and 90% (apex) cochlear positions from the basal end show a progression of HC differentiation from base to

*Figure 2 continued on next page*

Figure 2 continued

apex based on overall organization and stereociliary bundles in the hair cells (HCs). White brackets indicate one row of IHCs and three or more rows of OHCs. Magnified images show kinocilia of OHCs (Ca, Cb, Ea; yellow arrows) and primary cilia of Deiters' cells (Ca; white arrows) and pillar cells (Cb; white arrowheads). (F–J) In *Ift88* cKO mutants, cochlea shows a severe shortening (F), premature HC differentiation (G–I), and multiple extra rows of OHCs in the apex (H, I). Both kinocilia and primary cilia are absent from most HCs and supporting cells (SCs) (Ga, Gb; asterisks). Ectopic vestibule-like HCs are present in the Kölliker's organ (J). (K–O) In *Tbc1d32*$^{bromi}$ mutants, cochlea shows a severe shortening (K), premature HC differentiation (L–N), and multiple rows of OHCs in the apex (M, N). Kinocilia of HCs are present (La; yellow arrow) but primary cilia in the SCs are missing (La, Lb; asterisks). Ectopic vestibule-like HCs are present adjacent to inner HCs (M; yellow arrowheads) and in the Kölliker's organ (O). (P–S) In *Cilk1* KO mutants, cochlea shows a slight shortening (P), slightly premature HC differentiation (Q–S), and increased OHCs in the apex (S). Both kinocilia and primary cilia are abnormally elongated (Qa, Qb) and primary cilia are often absent from SCs (Qa; asterisk). Scale bar in B, 500 µm, also applies to F, K, P; scale bar in C, 10 µm, also applies to D, E, G–I, L–N, Q–S; scale bar in Ca, 2 µm, also applies to Cb, Ea, Ga, Gb, La, Lb, Qa, Qb; scale bar in J, 2 µm, also applies to O. (T–V) Quantification of ciliary lengths (T; measured from at least 100 cells from three or more embryos for each genotype), cochlear lengths (U; *n* = 5–7 embryos for each genotype), and HC numbers (V; *n* = 3–5 embryos for each genotype). Values and error bars represent the mean ± standard deviation. Statistical comparisons were made using the two-tailed Student's *t*-test (**p<0.01, ***p<0.001).

The online version of this article includes the following source data and figure supplement(s) for figure 2:

**Source data 1.** Raw data for cilia length, cochlear length, and HC number shown in *Figure 2*.
**Figure supplement 1.** SEM and immunofluorescent images of hair cells (HCs) of *Tbc1d32*$^{bromi}$ mutants.
**Figure supplement 1—source data 1.** Raw data for quantification of abnormal cilia shown in *Figure 2—figure supplement 1*.
**Figure supplement 2.** Comparison of hair cells (HCs) located at the same distance from the basal end of wild-type and ciliary mutant cochlea.
**Figure supplement 3.** Premature hair cell differentiation in ciliary mutant cochlea.

and 75% of the length of wild-type cochlea, respectively (*Figure 2—figure supplement 2A*). Interestingly, when comparing HCs located at the same distance, the differentiation status of the apical HCs of ciliary mutants looked similar to (*Cilk1* mutants) or mildly more mature (*Tbc1d32*$^{bromi}$ and *Ift88* mutants) compared to wild types (*Figure 2—figure supplement 2B–E*). These results may suggest that HC differentiation proceeds faster in ciliary mutants due to the shorter length of the cochlea.

Premature HC differentiation in ciliary mutants was further confirmed by examining expression patterns of genes essential for HC differentiation such as *Atoh1* and *Pou4f3*. In E17.5 wild-type cochlea, *Atoh1* and *Pou4f3* were expressed from the base to mid-apex, but not in the apex (*Figure 2—figure supplement 3A–B*; asterisks). In contrast, both genes were expressed in the entire duct, including the apex in the ciliary mutant cochleae (*Figure 2—figure supplement 3C–H*; red arrows), confirming that HC differentiation proceeds faster in ciliary mutants than in wild-type controls.

These analyses revealed that there are several common cochlear phenotypes among the three ciliary mutants (*Table 1*). First, all three mutants showed defective cochlear elongation with varying severity, with *Tbc1d32*$^{bromi}$ and *Ift88* cKO more severe than *Cilk1* KO mutants (*Figure 2U*). Second, total HC number decreased in proportion to the severity of cochlear shortening (*Figure 2V*). Third, there were multiple rows of HCs in the apical cochlea, with *Tbc1d32*$^{bromi}$ and *Ift88* cKO mutants more severe than *Cilk1* KO mutants (*Figure 2*), which could be related to the shortened cochlea. Fourth, all three mutant cochleae showed premature HC differentiation, with *Tbc1d32*$^{bromi}$ and *Ift88* cKO mutants more severe than *Cilk1* KO mutants (*Figure 2*). Fifth, ectopic vestibule-like HCs were present in the non-sensory region of the Kölliker's organ in *Tbc1d32*$^{bromi}$ and *Ift88* cKO mutants (*Figure 2J,O*). These common cochlear phenotypes of ciliary mutants closely resembled phenotypes observed in mutants with impaired SHH signaling (*Table 1*; *Bok et al., 2005*; *Bok et al., 2007*; *Bok et al., 2013*; *Brown and Epstein, 2011*; *Driver et al., 2008*; *Riccomagno et al., 2002*; *Tateya et al., 2013*), suggesting that most cochlear defects observed in ciliary mutants may be due to abnormal SHH signaling.

## SHH signaling is impaired in the developing cochlea of ciliary mutants

We examined whether SHH signaling is impaired in ciliary mutant cochleae at E14.5, an age when SHH signaling is active with an apical-to-basal gradient and regulates various essential cochlear developmental processes (*Bok et al., 2013*; *Liu et al., 2010*). SHH signaling was determined by analyzing SHH target genes, including *Ptch1* and *Gli1*, by in situ hybridization and quantitative real-time PCR (qPCR) (*Figure 3*). To visualize expression gradients along the cochlea, we semi-quantified the

**Table 1.** Comparisons of cochlear phenotypes of ciliary mutants and sonic hedgehog (SHH) pathway mutants*.

| | Impaired SHH signaling | | | | | Defective primary cilia | | |
|---|---|---|---|---|---|---|---|---|
| | Loss of SHH from both midline and ganglion sources[1] | Defective GLI[2] | Loss of SHH from ganglion source[3] | Loss of SMO after cochlear specification[4] | All combined | No cilia (*Ift88* cKO) | Abnormal morphology (*Tbc1d32*^*bromi*) | Abnormal length (*Cilk1* KO) |
| Complete loss of cochlea | +++ | − | − | − | +++ | − | − | − |
| Shortened cochlea | n.a. | +++ | +++ | + | +++ | +++ | +++ | + |
| Reduced HC number | n.a. | n.d. | +++ | + | +++ | +++ | +++ | + |
| Premature HC differentiation | n.a. | n.d. | +++ | + | +++ | +++ | +++ | + |
| Multiple rows of HCs in the apex | n.a. | +++ | +++ | − | +++ | +++ | +++ | − |
| Ectopic vestibule-like HCs | n.a. | + | n.d. | n.d. | + | + | + | − |
| Reversed HC differentiation wave | n.a. | n.d. | +++ | − | +++ | +++ | +++ | |
| Defective apical identity | n.a. | + | − | n.d. | + | + | + | + |
| Low-frequency hearing loss | n.a. | +[#] | n.d. | + | + | n.a. | n.a. | + |
| Impaired SHH signaling | +++ | +++ | +++ | n.d. | +++ | +++ | +++ | ++ |
| Defective hair bundle polarity | n.a. | − | − | − | − | +++ | − | − |

*This table summarizes cochlear phenotypes of ciliary mutant mice (this study) and SHH signaling mutant mice (previous reports).

+sign indicates the presence of defective phenotype. '+++" indicates that the phenotype is severe, '++" intermediate, and '+" mild.

−sign indicates the absence of defective phenotypes.

n.a. not applicable due to embryonic lethality or structural loss.

n.d. not determined.

[1](**Bok et al., 2007**; **Brown and Epstein, 2011**; **Riccomagno et al., 2002**).

[2](**Bok et al., 2007**; **Driver et al., 2008**).

[3](**Bok et al., 2013**; **Son et al., 2015**).

[4](**Tateya et al., 2013**).

[#]Hearing data are obtained from human patients with Pallister–Hall syndrome (**Driver et al., 2008**).

intensity of in situ hybridization signals in all cochlear sections (*Figure 3—figure supplement 1*). The strongest intensity among all wild-type sections was set to 100%, and the relative intensity of each section (Y-axis) was plotted according to the section number starting from the base to the apex (X-axis). Consistent with the shortened cochlear lengths (*Figure 2*), the number of cochlear sections of ciliary mutants was much less than that of wild-type controls (*Figure 3—figure supplement 1*).

In wild-type controls, direct targets of SHH signaling such as *Ptch1* and *Gli1* are expressed in a graded pattern, stronger in the apex and gradually weaker toward the base (*Figure 3A,B*). In *Ift88* cKO and *Tbc1d32*^*bromi* mutants, expression levels of *Ptch1* and *Gli1* were dramatically decreased in the shortened cochlea and lost the typical apical-to-basal gradients (*Figure 3E,F,J,K*; *Figure 3—figure supplement 1*). In *Cilk1* KO mutants, expression levels of *Ptch1* and *Gli1* were also reduced, but to a lesser extent than in *Tbc1d32*^*bromi* and *Ift88* cKO mutants (*Figure 3O,P*; *Figure 3—figure supplement 1*). Our qPCR results confirmed that expression levels of *Ptch1* and *Gli1* were significantly reduced in all three ciliary mutant cochleae, and *Tbc1d32*^*bromi* and *Ift88* cKO mutants showed a much greater reduction than *Cilk1* KO mutants (*Figure 3I,N,S*). These results indicate that SHH signaling is compromised at varying degrees in all three ciliary mutants.

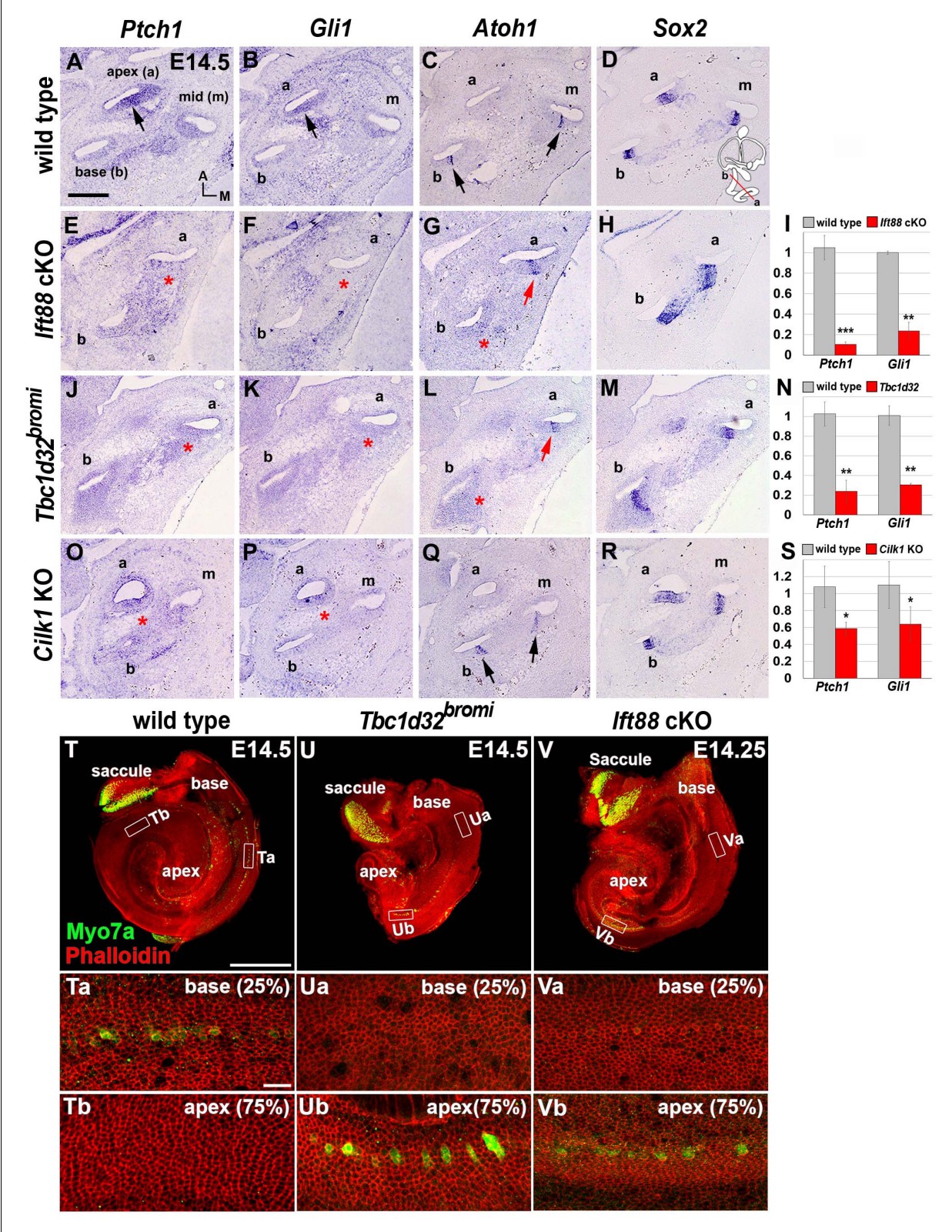

**Figure 3.** Impaired sonic hedgehog (SHH) signaling and reversed wave of HC differentiation in ciliary mutant cochleae. (**A–D**) In E14.5 wild-type cochlea, SHH target genes *Ptch1* and *Gli1* are expressed in a graded pattern of stronger in the apex (**A**, **B**; arrows) and weaker toward the base. *Atoh1* expression representing HC differentiation is observed in the base and middle cochlear turns but not in the apical turn (**C**; arrows), whereas *Sox2* expression representing prosensory domain is observed in all cochlear turns (**D**). (**E–H**) In *Ift88* cKO cochlea, *Ptch1* and *Gli1* are greatly downregulated

*Figure 3 continued on next page*

Figure 3 continued

(E, F; asterisks). *Atoh1* is ectopically expressed in the apex (G; red arrow) but not in the base (G; red asterisk). (J–M) In *Tbc1d32^bromi^* mutant cochlea, *Ptch1* and *Gli1* are greatly downregulated (J, K; asterisks). *Atoh1* is ectopically expressed in the apex (L; red arrow) but not in the base (L; red asterisk). (O–R) In *Cilk1* KO cochlea, *Ptch1* and *Gli1* are downregulated (O, P; asterisks). Unlike *Ift88* cKO and *Tbc1d32^bromi^* mutants, *Atoh1* is expressed in the base and middle turns (Q; arrows). (I, N, S) Quantitative real-time PCR analyses to determine expression levels of *Ptch1* and *Gli1* in the cochleae of *Ift88* cKO (n = 4), *Tbc1d32^bromi^* (n = 3), and *Cilk1* KO (n = 4) mutants relative to wild-type controls (n = 3). Values and error bars represent the mean ± standard error. Statistical comparisons were made using two-tailed Student's *t*-test (*p<0.05, **p<0.01). (T–V) E14.5 whole cochlear images stained with phalloidin and anti-MYO7A antibody. In wild-type controls, MYO7A immunofluorescence is detected in the basal cochlea region (T, Ta; 25% from basal end) but not in the apical region (Tb; 75% from basal end). In *Tbc1d32^bromi^* and *Ift88* cKO mutants, MYO7A immunofluorescence is detected in the apical cochlear region (Ub, Vb; 75% from basal end) but not in the basal region (Ua, Va; 25% from basal end). Scale bar in A, 100 μm, also applies to B–R; scale bar in T, 500 μm, also applies to U and V; scale bar in Ta, 20 μm, also applies to Tb, Ua, Ub, Va, and Vb.

The online version of this article includes the following source data and figure supplement(s) for figure 3:

Source data 1. Raw data for qRT-PCR assays shown in *Figure 3*.
Figure supplement 1. Relative signal intensity of in situ hybridization for *Ptch1*, *Gli1*, *Atoh1*, and *Sox2* along the cochlea.
Figure supplement 1—source data 1. Raw data for quantification of in situ hybridization signal intensity of *Ift88*cKO.
Figure supplement 1—source data 2. Raw data for quantification of in situ hybridization signal intensity of *Tbc1d32bromi*.
Figure supplement 1—source data 3. Raw data for quantification of in situ hybridization signal intensity of *Cilk1*KO.

## Defective ciliogenesis causes a reversal of HC differentiation wave

In the developing cochlea, the terminal mitosis of HC precursors starts from the apical region and spreads toward the basal direction, but HC differentiation occurs in an opposite wave from base to apex (*Chen et al., 2002*; *Lee et al., 2006*). Thus, HC precursors in the apex exit from the cell cycle first, but they are the last to differentiate. This unique basal-to-apical wave of HC differentiation is known to be controlled by an apex to base gradient of SHH signaling, which inhibits HC differentiation (*Bok et al., 2013*). We questioned whether the impaired SHH signaling observed in the ciliary mutants would disrupt this unique HC differentiation pattern. In E14.5 wild-type cochlea, the master regulator of HC differentiation, *Atoh1*, was expressed in *Sox2*-positive pro-sensory domains of the basal and middle, but not the apical turns, representing a basal-to-apical wave of HC differentiation (*Figure 3C,D*; *Chen et al., 2002*). Notably, in *Ift88* cKO and *Tbc1d32^bromi^* mutants, *Atoh1* was expressed in apical, but not basal cochlea (*Figure 3G,L*; red arrow and asterisk), suggesting a reversal of the HC differentiation wave. By contrast, *Atoh1* expression wave was not reversed, showing the typical basal-to-apical wave in *Cilk1* KO mutants (*Figure 3Q*). These results suggest that HC differentiation wave is affected differently depending on the degree of SHH defects in ciliary mutants.

The reversed apical-to-basal wave of HC differentiation in *Ift88* cKO and *Tbc1d32^bromi^* mutants was confirmed by myosin 7a (MYO7a) immunostaining to visualize differentiating HCs and phalloidin staining to visualize filamentous actin bundles (*Figure 3T–V*). In E14.5 wild-type controls, MYO7A immunoreactivity was observed in the basal cochlea (*Figure 3T* and Ta), but not in the apex (*Figure 3Tb*). In contrast, in *Tbc1d32^bromi^* (*Figure 3U–Ub*) and *Ift88* cKO (*Figure 3V–Vb*) mutants, MYO7A immunoreactivity was detected in the apical cochlea, but not in the base. Such reversal of the HC differentiation wave has only been reported in mutants in which SHH signaling is disrupted (*Bok et al., 2013*). Thus, we speculate that the reversal of HC differentiation wave observed in ciliary mutants is most likely due to dysregulation of SHH signaling caused by defective ciliogenesis.

## Ciliary frequency in the developing cochlea is correlated with the severity of cochlear defects

To better assess the correlation between deficits in SHH signaling and defects in primary cilia in the three mutants, we examined the mutant cochleae at E14.5, when SHH signaling is active in the cochlea (*Bok et al., 2013*; *Son et al., 2015*). In E14.5 wild-type cochlea, IHCs had begun differentiation at the base, based on the cortical actin condensation (*Figure 4B*; bracket and arrowheads), and primary cilia were present in almost all precursor cells in the cochlear epithelium, regardless of cell types and stages of differentiation (*Figure 4A–D,Q*). In contrast, in E14.5 *Ift88* cKO cochlea, primary cilia were absent due to the loss of IFT88, which is required for ciliogenesis (*Figure 4E–H,Q*), but cortical condensation was apparent in both IHCs and OHCs along the entire length of the cochlea (*Figure 4F–H*; brackets). Unexpectedly, in E14.5 *Tbc1d32^bromi^* cochlea, most precursor cells lacked primary cilia (*Figure 4I–L,Q*). This phenotype is different from E18.5 *Tbc1d32^bromi^* cochlea, which

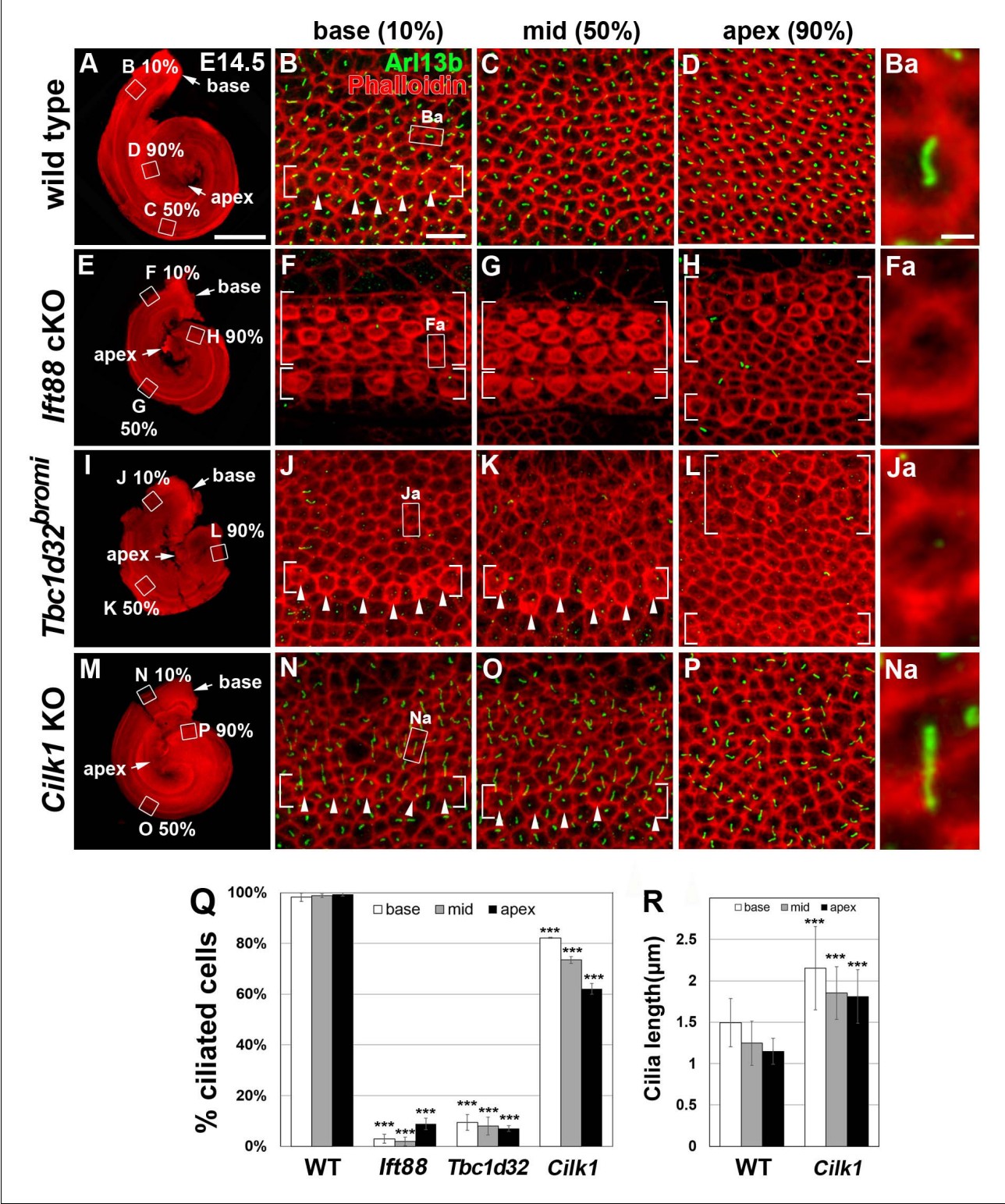

**Figure 4.** Frequency of ciliated cells in E14.75 ciliary mutant cochleae. Immunofluorescent staining of primarily cilia with anti-ARL13B antibody (green) and cell boundaries with phalloidin (red) of 14.5 cochleae from wild-type (A–D), *Ift88* cKO (E–H), *Tbc1d32^bromi* (I–L), and *Cilk1* KO (M–P) mice. Brackets indicate IHC and OHC rows, and arrowheads indicate cells with cortical actin condensation. Higher magnification images showing the lack of abnormally elongated cilia are displayed for each genotype (Ba, Fa, Ja, Na). Scale bar in A, 500 µm, also applies to E, I, and M; scale bar in B, 10 µm, also applies to C, D, F–H, J–L, and N–P. (Q) Percentages of ciliated cells are quantified in basal, middle, and apical regions of the cochlea from each genotype. The presence or absence of primary cilia is determined from at least 150 cells per region (at least 450 cells per embryo) from three different

*Figure 4 continued on next page*

*Figure 4 continued*

embryos for each genotype embryos. (R) Quantification of ciliary lengths measured from at least 100 cells from wild-type (n = 3) and *Cilk1* KO (n = 3) embryos. Values and error bars represent the mean ± standard deviation. Statistical comparisons were made using the two-way ANOVA with Bonferroni correction for multiple comparisons (**p<0.01, ***p<0.001).

The online version of this article includes the following source data for figure 4:

**Source data 1.** Raw data for cilia frequency and length shown in *Figure 4*.

showed a selective absence of primary cilia in the SCs only (*Figure 2L–N*). These results suggest that HCs acquire kinocilia, but not SCs, in *Tbc1d32^bromi* mutants at least by E18.5. Cortical condensation was present in both IHCs and OHCs in the apex (*Figure 4L*; brackets), but only in IHCs in the middle and base (*Figure 4J,K*; brackets and arrowheads), indicating a reversed apical-to-basal wave of HC differentiation. In E14.5 *Cilk1* KO cochlea, abnormally elongated primary cilia were present in 60–80% of the cells depending on the cochlear region (*Figure 4M–P,R*). These results show that *Ift88* cKO and *Tbc1d32^bromi* mutants, which are unciliated in most cochlear cells at E14.5, exhibit more severe cochlear defects than the ciliated *Cilk1* KO mutants. These results are consistent with the hypothesis that SHH mediates its effects via primary cilia.

## Hair bundle polarity is mildly affected in Tbc1d32^bromi and Cilk1 mutants, unlike Ift88 mutants

Defective PCP signaling is also associated with shortened cochlea and multiple rows of HCs in the apex (*Etheridge et al., 2008*; *Montcouquiol et al., 2003*). Accordingly, we examined whether PCP signaling is disrupted in the ciliary mutants by analyzing the basal body positioning as a proxy for hair bundle orientation, which is regulated by PCP signaling (*Jones et al., 2008*). Deviation of basal body positioning was determined by measuring the angle between a line connecting the basal body to the HC center and a line connecting the centers of three adjacent HCs (*Figure 5A*; *Copley et al., 2013*; *Yin et al., 2012*). When the basal body was positioned at the abneural (lateral) pole, the angle was defined as 0°. In E18.5 wild-type cochlea, basal body deviation was generally less than 30° in either clockwise or counterclockwise directions in IHCs and all three rows of OHCs (OHC1, OHC2, and OHC3) (*Figure 5F*). We thus considered a basal body deviation <30° as a normal range, although OHC3 often showed deviation >30° probably due to immaturity of apical and middle cochlear regions at this stage. *Figure 5F* illustrates the distribution of basal body angles from combined HCs in the base, middle, and apex of cochleae, whereas results from individual regions are shown in *Figure 5—figure supplement 1*.

In *Ift88* cKO mutants, basal body positioning was significantly deviated from controls especially in OHCs: 71.5% of OHCs showed deviations >30°, compared to 12.6% in wild-type controls, and deviations extend to ±180° (*Figure 5F*; *Figure 5—figure supplement 1*). These results confirmed the severe polarity defects reported in the *Ift88* cKO mutants (*Jones et al., 2008*). In contrast, only 23.0% of OHCs in *Tbc1d32^bromi* mutants exhibited deviations of >30° and 9.94% in *Cilk1* KO mutants (*Figure 5F*; *Figure 5—figure supplement 1*), and the deviations were seldom greater than 60°. These results indicate that unlike *Ift88* cKO, basal body positioning is only mildly affected in *Tbc1d32^bromi* and *Cilk1* KO mutants.

It was previously reported that *Cilk1* KO cochleae show greater deviation of kinocilia positioning in the middle but not basal turn (*Okamoto et al., 2017*). We also observed similar results, in which basal body deviations >30° in OHCs of *Cilk1* KO cochleae were more common in the middle (13.4%) than basal (5.8%) and apical (10.0%) regions (*Figure 5—figure supplement 1*). In *Tbc1d32^bromi* mutants, OHCs with deviations >30° were more common in all cochlear regions (base 23.8%, middle 22.0%, and apex 22.9%), compared to those in wild-type controls (base 2.9%, middle 19.0%, and apex 17.8%) (*Figure 5—figure supplement 1*). These results indicate that deviations of basal body positioning in *Cilk1* KO and *Tbc1d32^bromi* mutants are much milder than those of *Ift88* cKO mutants (base 71.0%, middle 75.4%, and apex 67.9%), suggesting that PCP defects based on kinocilia alignment are fairly mild in *Tbc1d32^bromi* and *Cilk1* KO mutants. Moreover, it has been reported that the localization of core PCP proteins is unaffected in the cochlea of ciliary mutants, including *Ift88* and *Cilk1* mutants (*Jones et al., 2008*; *Okamoto et al., 2017*). We also observed normal localization of FZD6 in *Tbc1d32^bromi* mutants (*Figure 2—figure supplement 1*). Together, these results

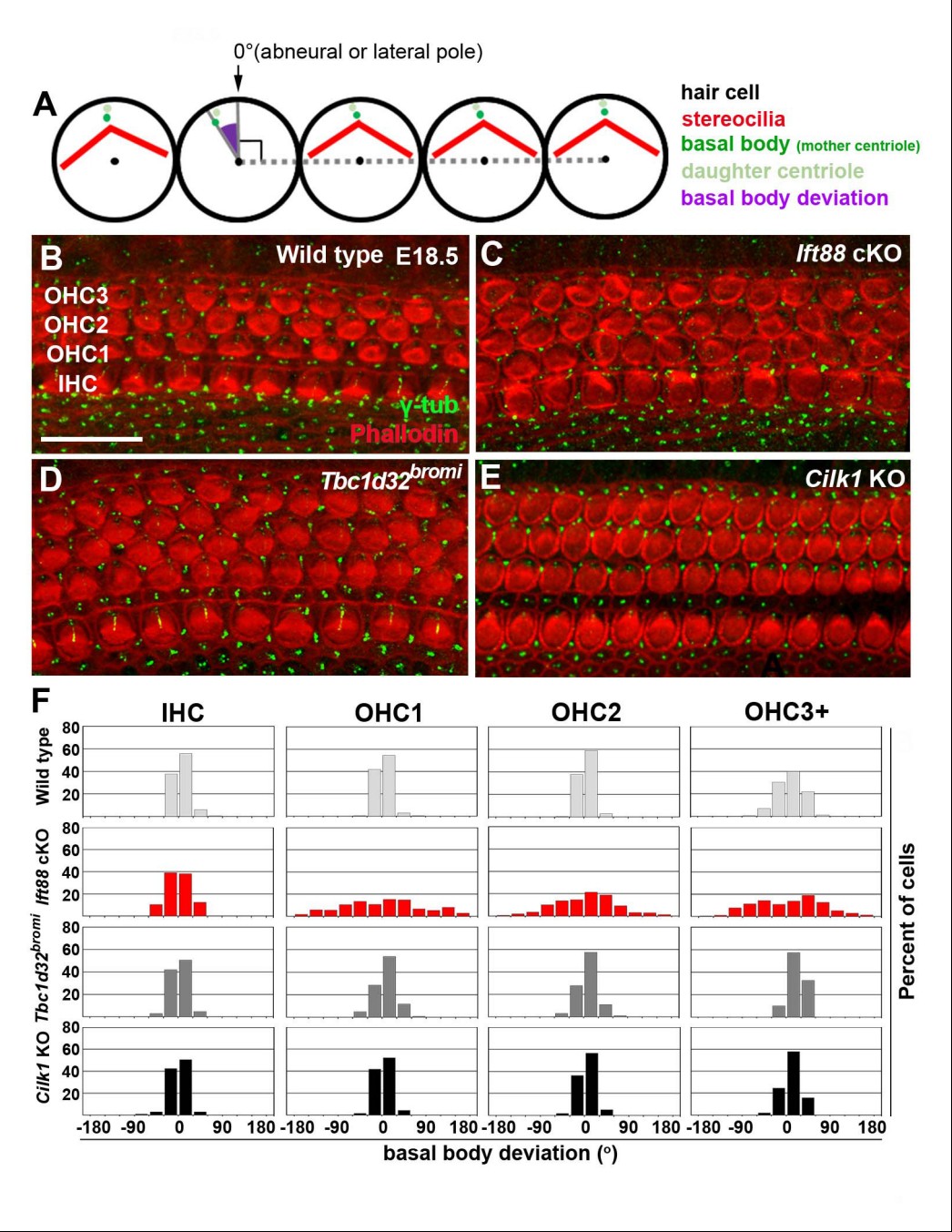

**Figure 5.** Deviation of basal body positioning in the three ciliary mutants. (**A**) Diagram showing how to measure the deviation of basal body position. When the basal body is positioned at the abneural side of HCs, the angle was defined as 0°. The basal position was determined by measuring angles deviated from the abneural side. (**B–E**) Whole mount cochlear images of WT and the three ciliary mutants stained with phalloidin to visualize F-actin (red) and anti-γ-tubulin antibody to visualize basal bodies (green). (**F**) Percentages of HCs showing the degrees of basal body deviation in each genotype. Angles are measured from at least 100 hair cells per embryo from at least three different embryos for each genotype. Scale bar in **B**, 20 μm, also applies to **C–E**.

The online version of this article includes the following source data and figure supplement(s) for figure 5:

**Source data 1.** Raw data for basal body positioning shown in *Figure 5*.
**Figure supplement 1.** Deviation of basal body positioning in *Tbc1d32^bromi* and *Cilk1* KO mutants.
**Figure supplement 1—source data 1.** Raw data for graphs shown in *Figure 5—figure supplement 1A-C*.
**Figure supplement 1—source data 2.** Raw data for graphs shown in *Figure 5—figure supplement 1D*.

suggest that PCP defects in the otic epithelium are unlikely to be a primary cause of cochlear shortening defects in ciliary mutants.

## Apical cochlear specification is affected in ciliary mutants

In addition to cochlear extension and HC differentiation, a gradient of SHH signaling plays a crucial role in specifying regional cochlear identity along the cochlea duct that prefigures tonotopic organization (*Son et al., 2015*), which based on differences in HC morphology and physiology, allows HCs at the basal cochlea to tune to high frequency sounds and those at the apex to lower frequencies. We wondered whether defective ciliogenesis leads to changes in regional cochlear identity. This was accomplished by analyzing expression patterns of genes, expressed in either a basal-to-apical or apical-to-basal gradient along the developing cochlea (*Figure 6A–P*; *Son et al., 2015*).

In wild-type controls, expression patterns of *A2m* and *Inhba* were strongest in the basal end and gradually weakened toward the apex (*Figure 6A,B,Q*), whereas expression patterns of *Msx1* and *Fst* were the opposite, being strongest in the apical end and gradually weaker toward the base (*Figure 6C,D,Q*; *Son et al., 2015*). In *Ift88* cKO mutants, while *A2m* and *Inhba* basal expression gradients were generally normal (*Figure 6E,F*, Qa, Qb), *Msx1* expression was greatly decreased and barely detectable in a few apical sections (*Figure 6G*; red asterisk, and Qa; arrow). *Fst* expression was also reduced in the mid-cochlea (*Figure 6H*; arrowhead) and its apical expression domain was reduced (*Figure 6Qb*; arrow). Similarly, *Tbc1d32^{bromi}* mutant cochlea showed relatively normal basal expression gradients of *A2m* and *Inhba* (*Figure 6I,J*, Qc, Qd), whereas apical expression domains of *Msx1* and *Fst* were decreased (*Figure 6K,L*; asterisk and arrowhead, and Qc, Qd). *Cilk1* KO mutants also showed decreases in apical expression gradients of *Msx1* and *Fst* to a lesser extent than *Ift88* cKO and *Tbc1d32^{bromi}* mutants (*Figure 6O,P*; red asterisk and arrowhead, and Qe, Qf), while there was no evidence of changes in basal expression gradients of *A2m* and *Inhba* (*Figure 6M,N*, Qe, Qf). Based on the changes in the expression gradients of these regional markers (*Figure 6Q*), all three ciliary mutants appeared to have apically truncated cochleae due to abnormal specification of apical cochlea, consistent with a deficit in SHH signaling (*Bok et al., 2007*; *Son et al., 2015*), which may account for the cochlear shortening in these ciliary mutants.

## Cilk1 mutants exhibit low frequency hearing loss

We next examined whether abnormal apical cochlear specification during development leads to tonotopic changes in the mature cochlea, resulting in low frequency hearing loss. Because all three ciliary mutants are embryonic lethal, we generated an inner ear-specific conditional knockout of the *Cilk1* gene by crossing *Foxg1^{Cre/+}* mice with *Cilk1^{fl/fl}* mice. The *Foxg1^{Cre/+}*; *Cilk1^{fl/fl}* mutant (*Cilk1* cKO) mice can survive until adulthood and reproduce the same cochlear phenotypes of *Cilk1* KO mutants, including abnormal elongation of the cilia, disruption of SHH signaling, shortening of the cochlea, and abnormal apical cochlear specification (*Figure 7—figure supplements 1*, *2,* and *3*). *Cilk1* cKO hearing function was examined at 4 weeks by measuring the thresholds of auditory brainstem responses (ABRs) in response to the click stimuli of broadband mixed sounds (*Figure 7A*) or pure tone sounds at individual frequencies (*Figure 7B–D*). ABR thresholds for click stimuli were slightly increased in *Cilk1* cKO mice, but the threshold shift was not statistically significant (*Figure 7A*). In contrast, ABR thresholds for pure tone stimuli were significantly elevated in lower frequencies ranging from 4 to 8 kHz but not in higher frequencies (*Figure 7B*), indicating a low frequency-specific hearing loss. The elevated ABR thresholds could be a result of a failure of IHCs to stimulate spiral ganglion neurons or OHCs to function as a cochlear amplifier, or both. The ABR wave I amplitudes, which reflect the summed responses of auditory nerve fibers, were not significantly different between wild-type and *Cilk1* cKO mice at low (8 kHz) or high (18 kHz) frequency sounds (*Figure 7C,D*). In contrast, the thresholds for distortion product otoacoustic emissions (DPOAEs), which indicate cochlear amplifier function of OHCs, were specifically elevated in response to lower frequencies at 6 and 8 kHz, but not to higher frequencies (*Figure 7E*). The input/output function of DPOAE amplitudes confirmed that DPOAE production was significantly decreased in response to a low frequency (8 kHz), but not to a high frequency (18 kHz), sound in *Cilk1* cKO mice (*Figure 7F,G*). Together, these results indicate that *Cilk1* cKO mutants suffer from low frequency-specific hearing loss, most likely due to defective OHC function in response to low frequencies.

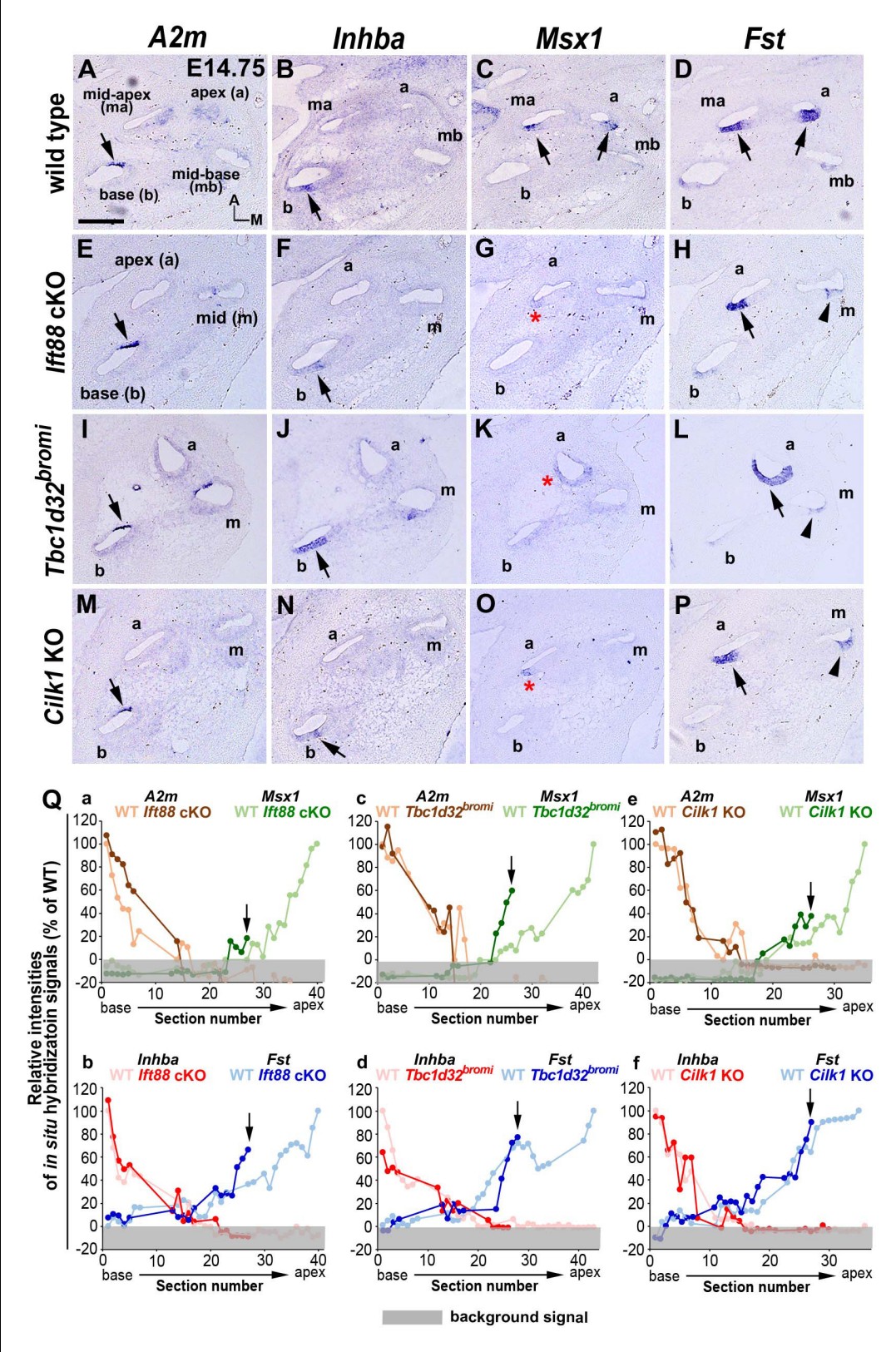

**Figure 6.** Specification of apical cochlear regional identity is compromised in ciliary mutants. (**A–P**) Gene expression patterns of basal cochlear markers (*A2m*, *Inhba*) and apical cochlear markers (*Msx1*, *Fst*) in cochleae of E14.75 ciliary mutants. (**A–D**) In wild-type controls, *A2m* and *Inhba* are expressed in the basal cochlear turn (**A**, **B**; *Figure 6 continued on next page*

*Figure 6 continued*

arrows), whereas *Msx1* and *Fst* are expressed in the apical cochlear turns (**C**, **D**; arrows). (**E–P**) In *Ift88* cKO, *Tbc1d32^{bromi}*, and *Cilk1* KO mutants, basal genes (*A2m* and *Inhba*) are generally unaffected and maintained in the basal cochlear turns (**E**, **F**, **I**, **J**, **M**, **N**; arrows). In contrast, *Msx1* is greatly downregulated in apical turns (**G**, **K**, **O**; red asterisks), and *Fst* is reduced and more restricted in the apical turns (**H**, **L**, **P**; black arrows for strong expression and arrowheads for weak expression). (**Q**) Relative signal intensity of in situ hybridization for *A2m, Inhba, Msx1,* and *Fst* along the cochlear duct of wild-type and ciliary mutants. For each gene, the strongest signal intensity among all wild-type cochlear sections is set to 100%, and the relative signal intensity (Y-axis) of each cochlear section of wild-type and ciliary mutants is plotted from the base to apex (X-axis). Arrows indicate the apical end of the shortened cochlear duct of each mutant. Gray boxes below 0% in each graph indicate background signals. Representative measurement graphs from one wild-type and one mutant cochlea for each gene are shown. Scale bar in **A**, 100 μm, also applies to **B–P**.

The online version of this article includes the following source data for figure 6:

**Source data 1.** Raw data for quantification of in situ hybridization signal intensity shown in *Figure 6Q*.

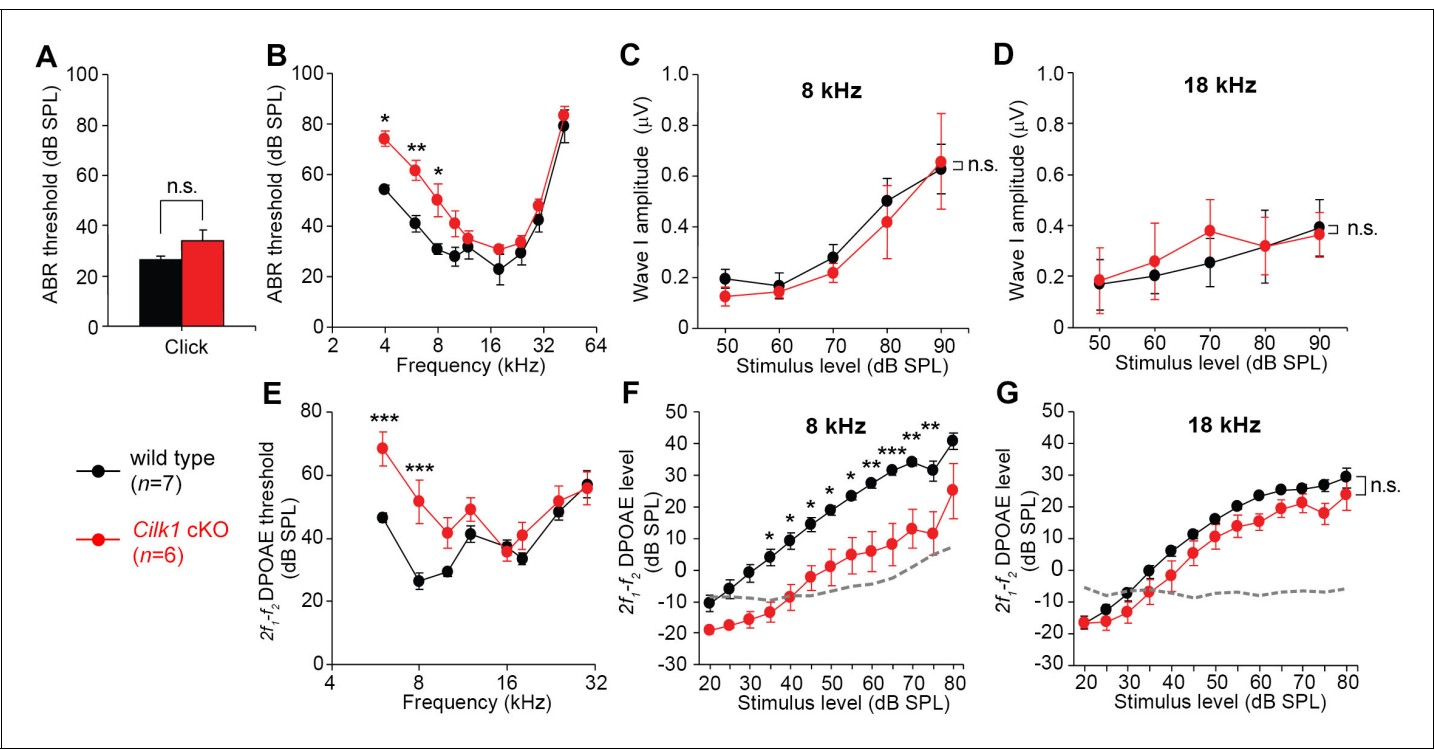

**Figure 7.** Low frequency hearing loss in 4-week-old *Cilk1* cKO mutants. (**A–D**) Auditory brainstem response (ABR) thresholds of wild-type and *Cilk1* cKO mice are not significantly different in response to click stimuli (**A**) but are significantly increased in low frequency pure tone stimuli at 4, 6, and 8 kHz but not in higher frequencies in *Cilk1* cKO mice (**B**). Input/output function analyses of wave I amplitudes show no significant differences between wild-type and *Cilk1* cKO mice at 8 and 18 kHz (**C**, **D**). (**E–G**) $2f_1$-$f_2$ distortion product otoacoustic emission (DPOAE) thresholds are significantly increased in low frequencies at 6 and 8 kHz but not in higher frequencies in *Cilk1* cKO mice (**E**). DPOAE input/output function analyses of $2f_1$-$f_2$ DPOAE levels show a significant reduction at 8 kHz (**F**) but not at 18 kHz (**G**) in *Cilk1* cKO mice. Values and error bars are mean ± standard error. Statistical comparisons were determined using the two-way ANOVA with Bonferroni correction for multiple comparisons (n.s., non-significant, *p<0.05, **p<0.01, ***p<0.001).

The online version of this article includes the following source data and figure supplement(s) for figure 7:

**Source data 1.** Raw data for ABR and DPOAE analyses shown in *Figure 7*.
**Figure supplement 1.** Cochlear phenotypes of *Cilk1* cKO mutants at E18.5.
**Figure supplement 1—source data 1.** Raw data for quantification of cochlear length and cilia length shown in *Figure 7—figure supplement 1*.
**Figure supplement 2.** Sonic hedgehog (SHH) signaling is impaired in *Cilk1* cKO mutant cochlea.
**Figure supplement 2—source data 1.** Raw data for qRT-PCR assays and quantification of in situ hybridization signal intensity shown in *Figure 7—figure supplement 2*.
**Figure supplement 3.** Specification of apical cochlear identity is impaired in *Cilk1* cKO mutants.
**Figure supplement 3—source data 1.** Raw data for quantification of in situ hybridization signal intensity shown in *Figure 7—figure supplement 3*.

## Hair bundle morphology is altered to adopt more basal cochlear properties in Cilk1 mutants

The low frequency hearing loss due to OHC dysfunction prompted us to examine OHC morphology in the mature cochlea of 4-week-old *Cilk1* cKO mutants. Scanning electron micrographs lacked evidence of IHC and OHC degeneration in *Cilk1* cKO cochlea (*Figure 8A–F*), indicating that the low frequency hearing loss is not due to HC loss in the apical cochlea. We thus examined whether the tonotopic properties of stereocilia were altered in 4-week-old *Cilk1* cKO cochlea. It is well known that stereocilia lengths are shorter at the base and gradually elongate toward the apex (*Wright, 1984*). We measured the stereocilia lengths of OHCs in wild-type and *Cilk1* cKO cochleae

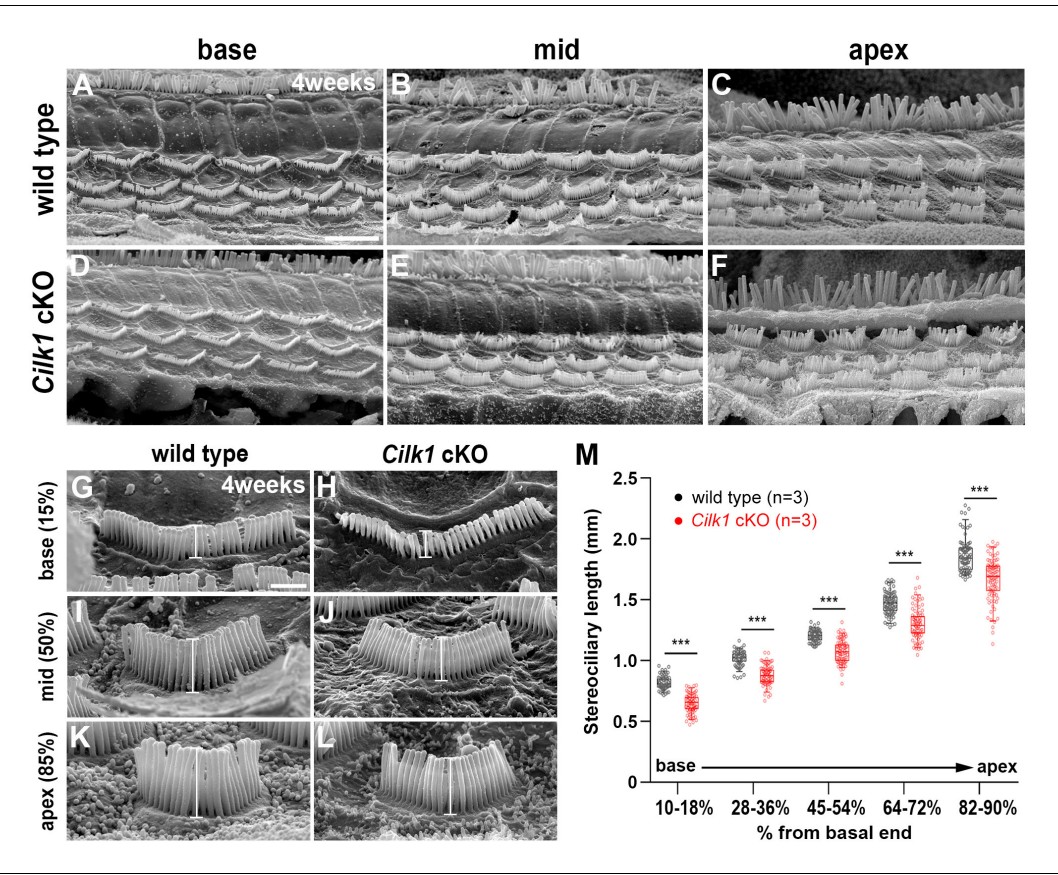

**Figure 8.** Decreased stereocilia lengths in 4-week-old *Cilk1* cKO mutants. (A–L) Scanning electron micrographs of the organ of Corti of wild-type and *Cilk1* cKO mutants. There is no obvious HC degeneration in the base, middle, and apex of the cochlea (A–F). Higher magnification images show that OHC stereociliary lengths are decreased in the basal, middle, and apical cochlea of *Cilk1* cKO mutants (G–L; brackets). (M) Quantification of stereociliary lengths of OHCs demonstrate significant decreases along the cochlear duct position in *Cilk1* cKO mutants. Stereociliary lengths were measured from at least 30 OHCs per each region from three animals per each genotype. Data are displayed as box plots. Individual dots represent individual data values, the boxes represent a range of 25–75%, the horizontal lines in the boxes represent the median, the whiskers represent the 5% and 95% values, and the points outside the whiskers represent outliers. Scale bar in A, 5 µm, also applies to B–F; scale bar in G, 1 µm, also applies to H–L. ***p<0.001, as determined by two-way ANOVA with Bonferroni correction for multiple comparisons.

The online version of this article includes the following source data and figure supplement(s) for figure 8:

**Source data 1.** Raw data for quantification of stereociliary length shown in *Figure 8*.

**Figure supplement 1.** Comparison of OHC stereociliary lengths at the same distance from the basal end of wild-type and *Cilk1* cKO mutant cochlea.

**Figure supplement 1—source data 1.** Raw data for stereociliary length shown in *Figure 8—figure supplement 1*.

from five different tonotopic locations based on a percentage of the overall length. Wild-type cochlea showed an increasing gradient of stereocilia lengths from base to apex (*Figure 8G,I,K,M*). Interestingly, in *Cilk1* cKO cochlea, stereocilia lengths were significantly decreased across all cochlear regions while maintaining the increasing gradient (*Figure 8H,J,L,M*). Since the cochlear ducts were shorter in *Cilk1* cKO mutants, we compared stereociliary lengths at the same distance from the base in wild-type and *Cilk1* cKO mutants (*Figure 8—figure supplement 1*). Interestingly, stereociliary lengths in the apical regions of *Cilk1* cKO mutants were comparable to those in the middle or mid-apical region of wild-type controls, indicating a loss (or truncation) of the apical cochlea (*Figure 8—figure supplement 1*). These results are consistent with the downregulation of apical cochlear markers in ciliary mutants (*Figure 6*). These results suggest that the tonotopic gradient of stereocilia lengths is shifted to adopt more basal features due to a failure of apical cochlea specification in *Cilk1* cKO cochlea, resulting in low frequency-specific hearing loss (*Figure 7*).

## Discussion

In the current study, we investigated the role of primary cilia in the cochlea to provide insights into the mechanism of hearing loss in ciliopathies. In addition to the essential role of kinocilia in determining hair bundle polarity (*Jones et al., 2008*; *May-Simera et al., 2015*; *Sipe and Lu, 2011*), primary cilia are present in the inner ear primordium long before kinocilia emerge in differentiating HCs and control multiple cochlear developmental events by mediating SHH signaling (*Table 1*). Notably, ciliary mutants mouse models exhibit low frequency hearing loss resulting from abnormal apical cochlear specification due to impaired SHH signaling.

### Impaired SHH signaling accounts for most cochlear phenotypes of ciliary mutants

Research has shown that there are two distinct sources of SHH signaling contributing to inner ear development. SHH signaling from the ventral midline sources, the floor plate and notochord, specifies the dorsoventral otic identity that determines vestibular and cochlear fates and also confers the apicobasal cochlear identity that prefigures the tonotopic axis (*Bok et al., 2007*; *Brown and Epstein, 2011*; *Son et al., 2015*). In contrast, SHH signaling from the spiral ganglion neurons located in the apical cochlea promotes cochlear elongation and determines the timing and wave of HC differentiation (*Bok et al., 2013*; *Liu et al., 2010*; *Son et al., 2015*; *Tateya et al., 2013*). Both sources provide an SHH activity gradient that is higher in the ventral inner ear (or apical cochlea) and gradually lower toward the dorsal inner ear (or basal cochlea), and this is mediated by a balance between GLI activator and repressor activities (*Bok et al., 2007*; *Bok et al., 2013*; *Driver et al., 2008*; *Son et al., 2015*).

Thus, inner ear defects may differ depending on when and where SHH signaling becomes impaired (*Table 1*). Briefly, when SHH signaling is absent (*Shh$^{-/-}$*) or when otic epithelium cannot transduce SHH signaling at all (*Foxg1$^{Cre}$;Smo$^{fl/fl}$*), the cochlea is completely lost (*Bok et al., 2007*; *Brown and Epstein, 2011*; *Riccomagno et al., 2002*). When SHH signaling is impaired due to defective GLI activator/repressor functions (*Gli2$^{-/-}$;Gli3$^{-/-}$*and *Gli3$^{\Delta699/\Delta699}$*), the cochlea is present, although it is severely shortened without apical identity and multiple extra rows of HCs are found in the distal region. Additionally, ectopic vestibular-like HCs are found in the Kölliker's organ (*Bok et al., 2007*; *Driver et al., 2008*). When SHH signaling is absent from the spiral ganglion source (*Foxg1$^{Cre}$;Shh$^{fl/fl}$* and *Neurog1-Cre$^{ERT2}$; Shh$^{fl/fl}$*), the cochlea is also shortened with multiple rows of HCs in the apex, and HCs are differentiated prematurely in a reverse apical-to-basal direction (*Bok et al., 2013*). Lastly, when SHH signaling becomes impaired after cochlear specification (*Emx2$^{Cre}$;Smo$^{fl/fl}$*), mild cochlear shortening and premature HC differentiation in the developing cochlea are accompanied by low frequency hearing loss in the mature cochlea (*Tateya et al., 2013*).

Our analyses of ciliary mutants, summarized in *Table 1*, revealed that cochlear phenotypes induced by ciliary defects encompass all of the defects resulting from impaired SHH signaling described above, except for a complete loss of the cochlea observed in *Shh$^{-/-}$* mutants. Although primary cilia are not essential for cochlear fate specification, these results demonstrate that primary cilia are required to mediate a multitude of spatiotemporally distinct processes regulated by SHH signaling in the developing cochlea, which are crucial for building a mature cochlea with normal auditory function.

## Temporally distinct roles of primary cilia in the developing cochlea

In addition to SHH signaling, primary cilia have been shown to be associated with PCP signaling, which is important for hair bundle polarity and cochlear extension (*Jones et al., 2008*; *May-Simera et al., 2015*; *Wallingford, 2010*). However, the polarity defects are not always associated with cochlear extension phenotypes such as inner ears that lack *Bbs8*, *Tmem67*, or *Vangl2* (*Abdelhamed et al., 2015*; *Copley et al., 2013*; *May-Simera et al., 2015*). Instead, cochlear shortening in ciliary mutants appears to be tightly associated with SHH signaling defects (*Table 2*). For example, in ciliopathy models exhibiting both cochlear shortening and polarity defects, including *Ift88*, *Kif3a*, *Ift27*, *Cilk1*, and *Tbc1d32^bromi* mutants (*Figure 2*; *Jones et al., 2008*; *May-Simera et al., 2015*; *Sipe and Lu, 2011*), SHH signaling is impaired (*Table 2*; *Bangs and Anderson, 2017*; *Eguether et al., 2018*; *Huangfu et al., 2003*). In contrast, in ciliopathy models displaying polarity defects but normal cochlear length, including *Alms1*, *Gmap210*, *Ift25*, and *Tmem67* mutants (*Abdelhamed et al., 2015*; *Leightner et al., 2013*; *May-Simera et al., 2015*), SHH signaling is unaffected (*Table 2*; *Chen et al., 2017*; *Keady et al., 2012*; *Chen et al., 2017*; *Smits et al., 2010*). These results indicate that cochlear shortening is observed in ciliary mutants if SHH signaling is impaired regardless of polarity defects (*Table 2*), suggesting that primary cilia regulate cochlear extension and hair bundle polarity in a temporally and spatially distinct manner. Furthermore, our results suggest that during early cochlear development, primary cilia act as the signaling center for SHH signaling to control multiple developmental steps, including cochlear elongation (*Figures 2* and *3*). Later on, in differentiating HCs, the polarized localization of kinocilia, which reflects the role of the basal body as a microtubule organizing center, determines hair bundle polarity (*Jones et al., 2008*; *May-Simera et al., 2015*; *Sipe and Lu, 2011*).

## Distinct ciliogenesis mechanisms between primary cilia and kinocilia

An interesting observation in E18.5 *Tbc1d32^bromi* mutants was the selective loss of primary cilia in SCs but not kinocilia in HCs (*Figure 2*), suggesting that TBC1D32/BROMI proteins are necessary for ciliogenesis in primary cilia, but not for kinocilia. Consistent with this, primary cilia are absent from nearly all precursor cells of *Tbc1d32^bromi* mutant cochlea at E14.5, when HC differentiation has just begun and kinocilia have not yet appeared (*Figure 4*). A similar phenotype has been reported in *Tmem67* mutant cochlea, where primary cilia are selectively lost in SCs but not HCs (*Abdelhamed et al., 2015*). These results indicate that primary cilia in SCs and kinocilia in HCs utilize distinct ciliogenesis mechanisms, and these results further suggest that molecular constituents and ciliary function are different between primary cilia and kinocilia. This speculation is consistent with the distinct roles of primary cilia as SHH signaling center in the developing cochlea and of kinocilia as a regulator of hair bundle polarity in differentiating HCs.

## Possible etiology of hearing loss in ciliopathy models

Based on comparative analysis of our results and previous studies (*Table 2*), we propose that cochlear phenotypes of ciliopathy mutants can be categorized into two groups: a group associated with impaired SHH signaling and the other group without SHH-associated defects.

The first group with defective ciliogenesis with abnormal SHH signaling includes most IFT complex mutants as well as *Cilk1* and *Tbc1d32^bromi* mutants (*Table 2*). We and others have observed that *Cilk1* cKO mutants, which show relatively mild SHH impairment and survive postnatally, exhibit low frequency hearing loss at 4–7 weeks (*Figures 7* and *8*; *Okamoto et al., 2017*). A previous study proposed that hair bundle polarity defects are a major cause of low frequency hearing loss in *Cilk1* cKO mutants (*Okamoto et al., 2017*). However, *Cilk1* cKO cochlea exhibited only a mild hair bundle polarity defect (*Figure 5*). Furthermore, polarity defects do not always result in hearing loss as in the case of *Bbs8^-/-* and *Alms1^-/-* mutants (see below), questioning the direct contribution of polarity defects to hearing function (*Jagger et al., 2011*; *May-Simera et al., 2015*). Our findings here suggest that apical cochlear specification is compromised in *Cilk1* cKO mutants due to impaired SHH signaling, resulting in basally shifted tonotopy and low frequency hearing loss. Consistent with this, the importance of SHH signaling for low frequency hearing in humans has been demonstrated in Pallister–Hall syndrome patients, in which SHH activity is impaired due to abnormal GLI activator function (*Driver et al., 2008*).

**Table 2.** Cochlear phenotypes of ciliary mutant mice*.

| | Mutants | Primary cilia (kinocilia) in the cochlea | Impaired SHH signaling | Shortened cochlear duct | Multiple rows of HCs | Hair bundle polarity defect | Hair bundle morphology defect | Core PCP protein localization defect | Hearing loss (HL) | Refs. |
|---|---|---|---|---|---|---|---|---|---|---|
| Ciliopathy models with impaired SHH signaling | Ift88 | Absent | ++ | ++ | ++ | ++ | ++ | − (Vangl2/Fzd3) | n.a. | This study and *Jones et al., 2008*; *May-Simera et al., 2015* |
| | Kif3a | Absent | ++[#] | ++ | ++ | ++ | ++ | − (Dvl2/Fzd3) | n.a. | *Huangfu et al., 2003*; *Sipe and Lu, 2011* |
| | Ift20 | Absent | n.d. | ++ | ++ | ++ | ++ | + (Vangl2) | n.a. | *May-Simera et al., 2015* |
| | Ift27 | Present | ++[#] | + | ++ | − | + | − (Vangl2) | n.a. | *Eguether et al., 2018*; *May-Simera et al., 2015* |
| | Cilk1 (Ick) | Elongated | + | + | − | +/- | +/- | − (Vangl2) | HL (1 month) low freq. | This study and *Okamoto et al., 2017* |
| | Tbc1d32 (bromi) | Malformed (HC); absent(SC) | ++ | ++ | ++ | +/- | +/- | n.d. | n.a. | This study |
| | Mks1 | Present | ++[#] | n.d. | n.d. | + | + | n.d. | n.d. | *Cui et al., 2011 Weatherbee et al., 2009* |
| | Cep290 (Nphp6) | Elongated | ++[#] | n.d. | n.d. | − | − | n.d. | HL (3–4 months) all freq. | *Hynes et al., 2014*; *Rachel et al., 2012* |
| Ciliopathy models with normal SHH signaling (or unknown) | Alms1 | Present | −[#] | n.d. | n.d. | ++ | + | n.d. | Normal hearing (1 month); HL (6–8 month) | *Chen et al., 2017*; *Collin et al., 2005*; *Jagger et al., 2011* |
| | Gmap210 (Trip11) | Present | −[#] | − | − | − | + | − (Vangl2) | n.d. | *May-Simera et al., 2015*; *Smits et al., 2010* |
| | Tmem67 (Mks) | Present (HC); absent(SC) | −[#] | − | − | ++ | ++ | − (Vangl2) | n.d. | *Abdelhamed et al., 2015*; *Lee et al., 2017*; *Leightner et al., 2013* |
| | Ift25 (Hspb11) | Present | +(E9.5); −(E10.5)[#] | − | − | − | − | − (Vangl2) | n.a. | *Keady et al., 2012*; *May-Simera et al., 2015* |
| | Bbs8 (Ttc8) | Present | n.d. | − | n.d. | ++ | ++ | + (Vangl2/Gαi3) | Normal hearing (2 months) | *May-Simera et al., 2015* |
| | Bbs4 | n.d. | n.d. | n.d. | n.d. | + | + | n.d. | n.d. | *Ross et al., 2005* |
| | Bbs6 (Mkks) | Present or absent | n.d. | n.d. | n.d. | + | + | − (Vangl2); + (Gαi3) | HL (3 months) in ~50% tested | *Ezan et al., 2013*; *May-Simera et al., 2015*; *Rachel et al., 2012*; *Ross et al., 2005* |

*This table summarizes cochlear phenotypes of ciliary mutant mice from this and previous studies.

[+]sign indicates the presence of defective phenotype. '+" and '++" indicate mild and severe phenotypes, respectively.

[−]sign indicates the absence of defective phenotype.

n.a. not applicable due to embryonic lethality.

n.d. not determined.

[#]SHH signaling data are obtained from other tissues (e.g. neural tube, limb) or in vitro assay systems.

In the second group of ciliopathy models, which include Bardet–Biedl syndrome (*Bbs4*[-/-], *Mkks*[-/-] (*Bbs6*[-/]), and *Bbs8*[-/]), Alström syndrome (*Alms1*[-/]), and Meckel–Gruber syndrome (*Tmem67*[-/]), the

primary cilia are present with relatively normal SHH signaling and normal cochlear lengths (*Table 2*; *Abdelhamed et al., 2015*; *Ezan et al., 2013*; *Jagger et al., 2011*; *Leightner et al., 2013*; *May-Simera et al., 2015*; *Rachel et al., 2012*; *Ross et al., 2005*). Despite evident neonatal polarity defects, hearing loss only develops after 6 months in *Bbs8*$^{-/-}$ and *Alms1*$^{-/-}$ mutants (*Table 2*; *Collin et al., 2005*; *Jagger et al., 2011*; *May-Simera et al., 2015*), and this hearing loss is associated with HC degeneration in *Alms1*$^{-/-}$ mice (*Collin et al., 2005*; *Jagger et al., 2011*). Histopathological analysis of human inner ears from Alström syndrome patients confirmed the degeneration of cochlear HCs, spiral ganglion neurons, and stria vascularis (*Nadol et al., 2015*). These results suggest that hearing loss in the second ciliopathy group with normal SHH signaling is caused by degeneration of cochlear cells and tissues, which could be due to dysfunction of SHH-independent ciliary roles or cilium-independent roles of BBS or ALSM1 proteins.

In the current study, we show that primary cilia play multiple essential roles in cochlear development and hearing function (*Table 1*). Comparative analyses of various ciliopathy models (*Table 2*) indicate that hair bundle polarity defects, although observed in most ciliopathy models, are unlikely to be the major cause of hearing loss. Instead, cochlear hypoplasia (shortening) and abnormal tonotopic organization should be considered as a potential etiology for hearing loss in a group of ciliopathies with defective ciliogenesis leading to impaired SHH signaling.

# Materials and methods

## Key resources table

| Reagent type (species) or resource | Designation | Source or reference | Identifiers | Additional information |
|---|---|---|---|---|
| Strain, strain background (*M. musculus*) | *Ift88*$^{fl}$ (B6.129P2-*Ift88*$^{tm1Bky}$/J) | The Jackson Laboratory | 022409; RRID:IMSR_JAX:022409 | |
| Genetic reagent (*M. musculus*) | *Cilk1*$^{tm1a/tm1a}$ | *Moon et al., 2014* | | |
| Genetic reagent (*M. musculus*) | *Tbc1d32*$^{bromi}$ | *Ko et al., 2010* | | |
| Genetic reagent (*M. musculus*) | *Pax2-Cre* (Tg(Pax2-cre)1Akg/Mmnc) | MMRRC | 10569; RRID:MMRRC_010569-UNC | |
| Genetic reagent (*M. musculus*) | *Foxg1-Cre* (129(Cg)-*Foxg1*$^{tm1(cre)Skm}$/J) | The Jackson Laboratory | 004337; RRID:IMSR_JAX:004337 | |
| Genetic reagent (*M. musculus*) | B6;SJL-Tg (ACTFLPe)9205Dym/J | The Jackson Laboratory | 003800; RRID:IMSR_JAX:003800 | |
| Antibody | Anti-ARL13B (Rabbit polyclonal) | This paper | | 1:3000 |
| Antibody | Anti-MyoVIIa (Rabbit polyclonal) | Proteus Biosciences | 25–6790; RRID:AB_10015251 | 1:200 |
| Antibody | Anti-Frizzled6 (Goat polyclonal) | R and D Systems | AF1526; RRID:AB_354842 | 1:200 |
| Antibody | Anti-γ tubulin (Rabbit polyclonal) | Sigma Aldrich | T3195; RRID:AB_261651 | 1:500 |
| Antibody | Anti-Sox2 (Goat polyclonal) | Santa Cruz Biotech | sc-17320; RRID:AB_2286684 | 1:500 |
| F-actin probe | Alexa Fluor 568 Phalloidin | Invitrogen | A12380 | 1:80 |
| Chemical compound, drug | DAPI | Invitrogen | D1306; RRID:AB_2629482 | 1 μg/mL |
| Recombinant DNA reagent | *Pax2* (Plasmid) | *Morsli et al., 1999* | In situ hybridization probe | pBluescript II (KS) backbone |
| Recombinant DNA reagent | *Otx2* (Plasmid) | *Morsli et al., 1999* | In situ hybridization probe | pBluescript II (KS) backbone |

*Continued on next page*

*Continued*

| Reagent type (species) or resource | Designation | Source or reference | Identifiers | Additional information |
|---|---|---|---|---|
| Recombinant DNA reagent | *Ptch1*_243 (plasmid) | *Son et al., 2015* | In situ hybridization probe | pBluescript II (KS) backbone |
| Recombinant DNA reagent | *Ptch1*_617 (plasmid) | *Son et al., 2015* | In situ hybridization probe | pBluescript II (KS) backbone |
| Recombinant DNA reagent | *Gli1* (plasmid) | *Son et al., 2015* | In situ hybridization probe | pBluescript II (KS) backbone |
| Recombinant DNA reagent | *Atoh1* (plasmid) | *Bok et al., 2013* | In situ hybridization probe | pBluescript II (KS) backbone |
| Recombinant DNA reagent | *Sox2* (plasmid) | *Son et al., 2015* | In situ hybridization probe | pCRII-TOPO |
| Recombinant DNA reagent | *Msx1* (plasmid) | *Son et al., 2015* | In situ hybridization probe | pCRII-TOPO |
| Sequence-based reagent | *A2m*_F | *Son et al., 2015* | PCR primers for in situ hybridization probe | TCC AGT CCT CTC TGC AGC AT |
| Sequence-based reagent | *A2m*_R | *Son et al., 2015* | PCR primers for in situ hybridization probe | AGT GAG CCC TTT ACC GGT TT |
| Sequence-based reagent | *A2m*_T7 | *Son et al., 2015* | PCR primers for in situ hybridization probe | TAA TAC GAC TCA CTA TAG GGA GAC TTG GAT CTT GGC ATT CAC A |
| Sequence-based reagent | *Inhba*_F | This paper | PCR primers for in situ hybridization probe | CCA AGG AAG GCA GTG ACC TG |
| Sequence-based reagent | *Inhba*_R | This paper | PCR primers for in situ hybridization probe | TGG GTC TCA GCT TGG TGG GC |
| Sequence-based reagent | *Inhba*_T7 | This paper | PCR primers for in situ hybridization probe | TAA TAC GAC TCA CTA TAG GGA GAG GGG CTG TGA CCC CTC ATG C |
| Sequence-based reagent | *Fst*_F | *Son et al., 2012* | PCR primers for in situ hybridization probe | CCTCCTGCTGCTGCTACTCT |
| Sequence-based reagent | *Fst*_R | *Son et al., 2012* | PCR primers for in situ hybridization probe | GCAGCGGGGTTTATTCTTCT |
| Sequence-based reagent | *Fst*_T7 | *Son et al., 2012* | PCR primers for in situ hybridization probe | TAATACGACTCACTATAGGGAGA TTCTTCTTGTTCATTCGACATTTT |
| Sequence-based reagent | *Pou4f3*_F | This paper | PCR primers for in situ hybridization probe | CGACTTACTTGAGCACATCTCG |
| Sequence-based reagent | *Pou4f3*_R | This paper | PCR primers for in situ hybridization probe | TTAGGCTCTCCAGGCTCCTC |
| Sequence-based reagent | *Pou4f3*_T7 | This paper | PCR primers for in situ hybridization probe | TAATACGACTCACTATAGGG AGAGAACCAGACCCTCACCACAT |
| Sequence-based reagent | *Gfi1*_F | This paper | PCR primers for in situ hybridization probe | TCTGCTCATTCACTCGGACA |
| Sequence-based reagent | *Gfi1*_R | This paper | PCR primers for in situ hybridization probe | TCCACAGCTTCACCTCCTCT |
| Sequence-based reagent | *Gfi1*_T7 | This paper | PCR primers for in situ hybridization probe | TAATACGACTCACTATAGGGAGA CAGCAGTCTCCCCAGAAGAG |
| Sequence-based reagent | *Ptch1*_F | PrimerBank | qPCR primers : 6679519a1 | AAAGAACTGCGGCAAGTTTTTG |
| Sequence-based reagent | *Ptch1*_R | PrimerBank | qPCR primers : 6679519a1 | CTTCTCCTATCTTCTGACGGGT |
| Sequence-based reagent | *Gli1*_F | PrimerBank | qPCR primers : 6754002a1 | CCAAGCCAACTTTATGTCAGGG |
| Sequence-based reagent | *Gli1*_R | PrimerBank | qPCR primers : 6754002a1 | AGCCCGCTTCTTTGTTAATTTGA |

*Continued on next page*

*Continued*

| Reagent type (species) or resource | Designation | Source or reference | Identifiers | Additional information |
|---|---|---|---|---|
| Sequence-based reagent | *Pax2-Cre_*F | MMRRC | Genotyping primer for ID # 10569 | GCC TGC ATT ACC GGT CGA TGC AAC GA |
| Sequence-based reagent | *Pax2-Cre_*R | MMRRC | Genotyping primer for ID # 10569 | GTG GCA GAT GGC GCG GCA ACA CCA TT |
| Sequence-based reagent | *Ift88 lox_*F | The Jackson laboratory | Genotyping primer for ID #16967 | GAC CAC CTT TTT AGC CTC CTG |
| Sequence-based reagent | *Ift88 lox_*R | The Jackson laboratory | Genotyping primer for ID #16969 | AGG GAA GGG ACT TAG GAA TGA |
| Sequence-based reagent | *Cilk1^{tm1a}_*F | *Moon et al., 2014* | Genotyping primer | CGC GTC GAG AAG TTC CTA TT |
| Sequence-based reagent | *Cilk1^{tm1a}_*R | *Moon et al., 2014* | Genotyping primer | ATC ATC CCG ATC AAG TCA GC |
| Sequence-based reagent | *Foxg1-Cre_*F | The Jackson Laboratory, | Genotyping primer for ID oIMR1084 | GCG GTC TGG CAG TAA AAA CTA TC |
| Sequence-based reagent | *Foxg1-Cre_*R | The Jackson Laboratory, | Genotyping primer for ID oIMR1085 | GTG AAA CAG CAT TGC TGT CAC TT |
| Sequence-based reagent | *Cilk1* WT_F | This paper | Genotyping primer | GCTGACTTGATCGGGATGAT |
| Sequence-based reagent | *Cilk1* WT_R | This paper | Genotyping primer | TGGCCAGGCTGGAACTCACTATA |
| Sequence-based reagent | *Cilk1 lox_*F | This paper | Genotyping primer | CACTGTGGGCAGAATCACCT |
| Sequence-based reagent | *Cilk1 lox_*R | This paper | Genotyping primer | CCGCCTACTGCGACTATAGA |
| Sequence-based reagent | *Tbc1d32/bromi_*F | *Ko et al., 2010* | Genotyping primer | AAT TTA GTC TCT GGG CAC AAC AA |
| Sequence-based reagent | *Tbc1d32/bromi_*R | *Ko et al., 2010* | Genotyping primer | TCA TCA GCT TTC ATA GCT TCA CA |
| Software, algorithm | SPSS | SPSS | RRID:SCR_002865 | |
| Software, algorithm | Graphpad Prism 8.0 | Graphpad | RRID:SCR_002798 | |

## Mice

Inner ear-specific *Ift88* conditional knockout (*Ift88* cKO) mice were generated by crossing *Ift88^{fl/fl}* mice (JAX) with *Pax2-Cre* transgenic mice (*Ohyama and Groves, 2004*). The generation of *Tbc1d32^{bromi}* mutant (*Ko et al., 2010*) and *Cilk1^{tm1a/tm1a}* knockout (*Cilk1* KO) mice (*Moon et al., 2014*) was previously described. *Cilk1^{fl/fl}* mice were generated by crossing *Cilk1^{tm1a/+}* heterozygote mice with transgenic mice expressing a FLP1 recombinase gene under the control of the ACTB promoter (B6;SJL-Tg(ACTFLPe)9205Dym/J; JAX). Inner ear-specific *Cilk1* conditional knockout (*Cilk1* cKO) mice were generated by crossing *Cilk1^{fl/fl}* mice with *Foxg1^{Cre}* mice (*Hébert and McConnell, 2000*). *Cilk1^{tm1a/+}* heterozygous mice, *Cilk1^{fl/fl}* mice, and *Ift88^{fl/fl}* mice were maintained in a C57BL/6N background and *Tbc1d32^{bromi}* mutant mice in a C57BL/6J background. All animal protocols were approved (No. 2018–0023) by the Institutional Animal Care and Use Committee at Yonsei University College of Medicine.

## Immunofluorescent staining and measurement of HC number and ciliary length and number

Immunofluorescent staining was performed as previously described (*Son et al., 2015*). Primary antibodies used were anti-ARL13B (1:3000, generated from rabbit) (*Moon et al., 2014*), anti-MYO7A (1:200; Proteus biosciences, 256790), anti-acetylated tubulin (1:200, Sigma, T7451), anti-γ-tubulin (1:500, Sigma, T3195), anti-FZD6 (1:200, R and D Systems, AF1526), and anti-SOX2 (1:500, Santa Cruz Biotechnology, dc-17320). Secondary antibodies used were Alexa Fluor 488 antibodies (1:200, Thermo Fisher Scientific, A11008, A11055, A11001) and Alexa Fluor 568 Phalloidin (1:100, Thermo

Fisher Scientific, A12380). Immunolabeled cochlear tissues were mounted with ProLong Gold Anti-fade Mountant (Thermo Fisher Scientific, P36930). All immunofluorescent images are representative of at least three different samples in two or more independent experiments, and all measurements were performed with at least three different samples.

## Basal body positioning

To analyze basal body positioning, the organ of Cori was immunolabeled with anti-γ-tubulin anti-body (γ-tubulin, 1:500, Sigma, T3195) and Alexa Fluor 568 Phalloidin (1:100, Thermo Fisher Scientific, A12380). The deviation of basal body position was determined by measuring the angle between a line connecting the basal body and the center of HC and a line connecting the centers of three adjacent HCs using ImageJ software (*Figure 5A*; *Copley et al., 2013*; *Yin et al., 2012*). When the basal body was located at the abneural pole of HCs, the angle was defined as 0° (*Figure 5A*). The angles were measured from at least 30 HCs in each row of IHC, OHC1, OHC2, and OHC3+ from each cochlear region (base, middle, and apex).

## In situ hybridization and measurement of signal intensities

Inner ears dissected from E14.5 embryos were fixed in 4% paraformaldehyde overnight and embedded in Tissue-Tek optimum cutting temperature (OCT) compound. Inner ear tissues were sectioned at a 12 µm thickness using a cryostat (Thermo Fisher Scientific). Serial cochlear sections were collected from the base to the apex onto Superfrost Plus microscope slides (Thermo Fisher Scientific) and subjected to in situ hybridization as previously described (*Son et al., 2015*). Antisense RNA probes for *Shh*, *Ptch1*, *Gli1*, *Otx2*, *Msx1*, *Atoh1*, *Sox2*, *Fst*, *A2m*, and *Inhba* were prepared as previously described (*Ankamreddy et al., 2019*; *Son et al., 2015*; *Son et al., 2012*). Images of in situ hybridization were acquired using a Leica DM2700 optical microscope.

To visualize expression gradients along the cochlear duct, in situ hybridization signal intensities were measured using Multi Gauge software (FUJIFILM). Images of all cochlear sections were collected and numbered from the base to the apex. The number of cochlear sections were around 35–42 in wild-type cochlea, and less for ciliary mutants, between 26 and 31. To measure signal intensities in cochlear section, a rectangle (15 × 4.5 µm for *A2m*, *Inhba*, *Msx1*, *Ptch1*, *Gli1*, *Atoh1*, and *Sox2* and 20 × 6 µm for *Fst*) was positioned on the cochlear epithelial region that shows hybridization signals for the gene of interest (e.g. the lesser epithelial ridge area for *Msx1*), and the signal intensity within the rectangle was measured as a quantum level (Q). Then, the same size rectangle was positioned in a nearby epithelial region that is not expressing the gene of interest to measure background level (B). The intensity of in situ hybridization signal in each cochlear section was calculated by subtracting the background level (B) from the quantum level (Q) and designated as QB. Cochlear sections that were obliquely sectioned due to the coil structure, such as the curvature of the spiral, were excluded from the analysis. For each gene, the lowest QB value among all sections with positive signals from a cochlea was set as 'the threshold QB', and the relative signal intensity of each section was calculated as follows:

$$\% \,\text{intensity of a section} = \frac{(\text{QB value of a section}) - (\text{the threshold QB value})}{(\text{the highest QB value among all sections}) - (\text{the threshold QB value})}$$

The in situ hybridization signal with the highest QB value among all cochlear sections from a wild type was designated as 100% intensity and the signal with the threshold QB value as 0% intensity. The percent intensities of each section from wild-type and mutant cochleae (Y-axis) were plotted according to the section number starting from the base to the apex (X-axis). Each graph shown in figures is a representative result conducted on a single sample of wild type and mutant obtained from the same experiment.

## Scanning electron microscopy (SEM) and measurement of stereociliary lengths

Cochlear samples from 4-week-old *Cilk1* cKO mice were prepared for SEM analysis as previously described (*Son et al., 2015*). Platinum coated specimens were mounted on a stub holder and imaged using a cold field emission scanning electron microscope (JSM-7001F; JEOL). SEM images of HCs were taken from the lateral side so that the tallest stereocilia of each HC could be easily

visible for analysis. For measurements of stereociliary lengths along the cochlear duct, SEM images obtained from entire cochlear regions were combined using Adobe Photoshop software, and the cochlear regions were divided into five regions 10–18%, 28–36%, 46–54%, 64–72%, and 82–90% from the basal end to represent base, mid-base, mid, mid-apex, and apex of the cochlea, respectively. We excluded 0–10% (basal end) and 90–100% (apical end) regions from measurement because of irregular stereociliary morphology often found in these regions. The lengths of the three tallest stereocilia were measured from at least 30 HCs in each region from three wild-type and three *Cilk1* cKO mice using ImageJ software (NIH). Averages of the three tallest stereocilia in each HC were plotted as a box plot using Prism 8.0 software (GraphPad, USA). In the box plot, individual dots represent individual data values, the box represents a range of 25–75%, the horizontal lines in the boxes represent the median, the whiskers represent 5% and 95% values, and the points outside the whiskers represent outliers.

## Auditory brainstem response

ABR thresholds were measured in a sound-proof chamber using Tucker-Davis Technologies (TDT) RZ6 digital signal processing hardware and the BioSigRZ software package (Alachua, FL, USA). Subdermal needles (electrodes) were positioned at the vertex and ventrolateral to the right and left ear of the anesthetized mice. Calibrated click stimulus (10-μs duration) or tone burst stimuli (5 ms duration) at 4, 6, 8, 10, 12, 18, 24, 30, and 42 kHz were produced using the SigGenRZ software package and an RZ6 digital signal processor and delivered to the ear canal through a multi-field 1 (MF1) magnetic speaker (TDT). The stimulus intensity was increased from 10 to 95 dB SPL in 5 dB SPL increments. The ABR signals were fed into a low-impedance Medusa Biological Amplifier System (RA4LI, TDT), which then was used to deliver the signal to the RZ6 digital signal processing hardware. The recorded signals were filtered using a 0.5–1 kHz band-pass filter, and the ABR waveforms in response to 512 tone bursts were averaged. The ABR thresholds for each frequency were determined using the BioSigRZ software package. Wave I amplitudes were determined at two specific frequencies (8 and 18 kHz) by calculating P1-N1 peak amplitudes (in μV) as input/output (I/O) functions with stimulus levels increased from 50 to 90 dB SPL in 10 dB SPL increments.

## Distortion product otoacoustic emission

DPOAEs were measured using a combined TDT microphone-speaker system. Primary stimulus tones were produced using an RZ6 digital signal processor and the SigGenRZ software package and delivered through a custom probe containing an ER 10B+ microphone (Etymotic, Elk Grove Village, IL, USA) with MF1 speakers positioned in the ear canal. The primary tones were set at a frequency ratio ($f_2/f_1$) of 1.2 with target frequencies at 6, 8, 10, 12, 16, 18, 24, and 30 kHz. The $f_1$ and $f_2$ intensities were set at equal levels (L1 = L2) and increased from 20 to 80 dB SPL in 5 dB SPL increments. The resultant sounds in response to the primary tones were received by the ER 10B+ microphone in the custom probe and recorded using the RZ6 digital signal processor. The input/output (I/O) functions for the DPOAEs were determined at two specific frequencies (8 and 18 kHz). The intensity levels of the primary tones were increased from 20 to 80 dB SPL in 5 dB SPL increments. At each primary tone for each intensity for the I/O functions, the fast Fourier transform (FFT) was performed using the BioSigRZ software package to determine the average spectra of the two primaries, the $2f_1$-$f_2$ distortion products, and the noise floors.

## Statistical analysis

Statistical comparisons were made using two-way analysis of variance (ANOVA) with Bonferroni corrections for multiple comparisons for ABRs, DPOAEs, and stereociliary length, and with Student's t-tests (two-tailed) for ciliary length, cochlear length, HC number, and ciliary frequency using Prism 8.0 (GraphPad, San Diego, CA, USA) and SPSS (IBM Institute). All graphs with statistical analysis are expressed as the mean ± standard error. Statistical significance is indicated in the figures as n.s., non-significant ($p > 0.05$), *$p < 0.05$, **$p < 0.01$, ***$p < 0.001$.

## Acknowledgements

We thank Dr. Doris Wu for critical reading of the manuscript, Dr. Ping Chen and Dr. Sun Myoung Kim for providing *Ift88* cKO embryo samples for initial study, and members of the Bok laboratory for constructive discussion of the manuscript.

## Additional information

### Funding

| Funder | Grant reference number | Author |
|---|---|---|
| National Research Foundation of Korea | NRF-2014M3A9D5A01073865 | Jinwoong Bok |
| National Research Foundation of Korea | NRF-2016R1A5A2008630 | Jinwoong Bok |
| National Research Foundation of Korea | NRF-2017R1A2B3009133 | Jinwoong Bok |
| National Research Foundation of Korea | NRF-2014M3A9D5A01073969 | Hyuk Wan Ko |

The funders had no role in study design, data collection and interpretation, or the decision to submit the work for publication.

### Author contributions

Kyeong-Hye Moon, Conceptualization, Investigation, Visualization, Formal analysis, Methodology, Validation, Writing—original draft, Writing—review and editing; Ji-Hyun Ma, Validation, Investigation; Hyehyun Min, Formal analysis, Validation; Heiyeun Koo, Validation, Investigation, Methodology; HongKyung Kim, Investigation; Hyuk Wan Ko, Jinwoong Bok, Conceptualization, Supervision, Funding acquisition, Writing - review and editing

### Author ORCIDs

Kyeong-Hye Moon (ID) https://orcid.org/0000-0002-6015-1690
Jinwoong Bok (ID) https://orcid.org/0000-0003-1958-1872

### Ethics

Animal experimentation: All animal protocols were approved by the Institutional Animal Care and Use Committee (IACUC) at Yonsei University College of Medicine with NIH guidelines (No. 2018-0023).

### Decision letter and Author response

Decision letter https://doi.org/10.7554/eLife.56551.sa1
Author response https://doi.org/10.7554/eLife.56551.sa2

## Additional files

### Supplementary files

• Transparent reporting form

### Data availability

All data generated or analysed during this study are included in the manuscript. Source data files have been provided for all the data that are represented as graphs in Figures.

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
