## [Decision Letter]

**Acceptance summary:**

Your work has helped demonstrate the role of cilia in cochlear duct development and patterning of the organ of Corti. This work potentially explains the etiology of hearing defects associated with a subset of ciliopathies, and elucidates the role of Sonic hedgehog ciliary signaling in inner ear development.

**Decision letter after peer review:**

Thank you for submitting your article "Dysregulation of sonic hedgehog signaling causes hearing loss in ciliopathy mouse models" for consideration by *eLife*. Your article has been reviewed by three peer reviewers, and the evaluation has been overseen by a Reviewing Editor and Kathryn Cheah as the Senior Editor. The reviewers have opted to remain anonymous.

The reviewers have discussed the reviews with one another and the Reviewing Editor has drafted this decision to help you prepare a revised submission.

Summary:

The reviewers appreciated the extensive amount of data, the interesting experimental system, and the importance of the biological question, but were concerned that the study was skewed towards a preconceived conclusion. They found that this study is biased in its presentation, in some cases ignoring or diminishing data that does not agree with the preconceived conclusion that the primary effects of a loss of cilia are all mediated through SHH signaling rather than, for example, defects in apicobasal polarity, planar cell polarity or some other consequence of ciliogenesis. The reviewers agreed that more data to support the role of SHH signaling dysregulation in the hearing loss phenotype and/or a rewriting of the paper to provide a more measured interpretation of the results is required before the manuscript can be considered for publication.

Essential revisions:

A number of conclusions are based on non-quantitative analyses. Several conclusions lack statistical support of any kind. For example, the paper would be improved by quantification of the data in Figure 3 (perhaps similar to the analysis used in Figure 6).

This paper does not show a direct link between low tone sensitivity loss and the SHH phenotype. Rather, the authors present a second Cre line, crossed with the least severely affected line (*Ick*), and show mild alterations to the cilia, without providing evidence that SHH is also disrupted in this ciliopathy model. Therefore, the functional conclusion is based on faith that the phenotype follows from the signaling alterations shown in the more severe models. Further, it is not clear what the consequences of *Foxg1-Cre Ick* cKO is: it is unclear if the tonal loss is due to abnormal development or due to the cilia phenotype. Further, it is unclear if SHH is meaningfully altered at all in those mutants, or if the cochleas are morphologically abnormal (e.g. truncated apex). A formal quantification of SHH in the *Foxg1-Cre* cKO mice is critical for the conclusion that SHH disruptions in this model are what cause low tone sensitivity loss.

---

## [Author Response]

Summary:The reviewers appreciated the extensive amount of data, the interesting experimental system, and the importance of the biological question, but were concerned that the study was skewed towards a preconceived conclusion. They found that this study is biased in its presentation, in some cases ignoring or diminishing data that does not agree with the preconceived conclusion that the primary effects of a loss of cilia are all mediated through SHH signaling rather than, for example, defects in apicobasal polarity, planar cell polarity or some other consequence of ciliogenesis. The reviewers agreed that more data to support the role of SHH signaling dysregulation in the hearing loss phenotype and/or a rewriting of the paper to provide a more measured interpretation of the results is required before the manuscript can be considered for publication.

We thank the editors and reviewers for providing such comprehensive and insightful comments that have helped strengthen our conclusions and greatly improved our manuscript. We understand the concerns from the reviewers that our study is biased towards a preconceived conclusion that SHH signaling dysregulation is a major cause of cochlear defects in ciliary mutants. This critical comment has spurred us to rethink all the results from a different perspective, conduct additional essential experiments, and rewrite parts of the manuscript. We believe that the conclusions of our study are now much clearer and better supported by experimental evidence.

Essential revisions:A number of conclusions are based on non-quantitative analyses. Several conclusions lack statistical support of any kind. For example, the paper would be improved by quantification of the data in Figure 3 (perhaps similar to the analysis used in Figure 6).

We understand the reviewers’ concern that some conclusions are based on non-quantitative analyses. In the revised manuscript, every effort has been made to include quantitative measurements and statistical analyses whenever results can be quantified, such as numbers, lengths, and hearing thresholds. We have also performed quantitative real-time PCR to confirm gene expression changes. Besides, as the reviewer suggested, we quantified in situ hybridization intensities for the data in Figure 3 and new in situ hybridization results for *Cilk1(Ick)* cKO mutants. Following is a list of new quantification data included in the revised manuscript.

– Figure 1—figure supplement 1: Quantification and statistical analyses for cilia numbers and in situ hybridization signal intensities for the data in Figure 1.

– Figure 2—figure supplement 1: Quantification and statistical analyses for frequency and diameter of swollen ciliary tips of *Tbc1d32^bromi^* mutants.

– Figure 3I, N, S: qPCR results for SHH target genes to support the data in Figure 3.

– Figure 3—figure supplement 1: Quantification for in situ signal intensities in Figure 3.

– Figure 4R: Quantification and statistical analyses for ciliary length for wild type and ciliary mutants at E14.5.

– Figure 5—figure supplement 1D: Statistical analyses for basal body deviation for the data in Figure 5 and Figure 5—figure supplement 1.

– Figure 7—figure supplement 1: Quantification and statistical analyses for cochlear length and ciliary length of *Cilk1(Ick)* cKO mutants.

– Figure 7—figure supplement 2 and Figure 7—figure supplement 3: Quantification for in situ hybridization intensities of *Cilk1(Ick)* cKO mutants.

This paper does not show a direct link between low tone sensitivity loss and the SHH phenotype. Rather, the authors present a second Cre line, crossed with the least severely affected line (Ick), and show mild alterations to the cilia, without providing evidence that SHH is also disrupted in this ciliopathy model. Therefore, the functional conclusion is based on faith that the phenotype follows from the signaling alterations shown in the more severe models. Further, it is not clear what the consequences of Foxg1-Cre Ick cKO is: it is unclear if the tonal loss is due to abnormal development or due to the cilia phenotype. Further, it is unclear if SHH is meaningfully altered at all in those mutants, or if the cochleas are morphologically abnormal (e.g. truncated apex). A formal quantification of SHH in the Foxg1-Cre cKO mice is critical for the conclusion that SHH disruptions in this model are what cause low tone sensitivity loss.

We thank the reviewer for pointing out the weak link between low tone sensitivity loss and SHH phenotype in *Foxg1-Cre; Cilk1(Ick)* cKO mice. We used this inner ear-specific *Cilk1(Ick)* conditional mutants to examine the effect of defective cilia on auditory function, because all three ciliary mutants used for developmental studies (*Ift88* cKO, *Tbc1d32^bromi^*, *Cilk1(Ick)* KO) were embryonic lethal. Fortunately, we found that the inner ear-specific deletion of *Cilk1* does not lead to an embryonic lethality, and we assumed that *Foxg1-Cre; Cilk1* cKO would cause the same cochlear phenotypes as *Cilk1(Ick)* whole body KO. Nevertheless, as the reviewer pointed out, it is important to confirm that *Foxg1-Cre; Cilk1* cKO mutants reproduce the cochlear phenotypes of *Cilk1* KO mutants, especially the reduction of SHH activity.

In the revised manuscript, we have analyzed the developmental cochlear phenotypes of *Foxg1-Cre; Cilk1* cKO mutants and observed following results:

– The lengths of primary cilia are abnormally elongated in the developing cochlea, and the cochlear duct is shortened (whole mount immunostaining results in Figure 7—figure supplement 1)

– SHH target genes such as *Ptch1* and *Gli1* are downregulated (in situ hybridization and their intensity measurements, and qPCR in Figure 7—figure supplement 2)

– Apical cochlear markers such as *Msx1* and *Fst* are downregulated, but not basal cochlear markers such as *A2m* and *Inhba* (in situ hybridization and their intensity measurements in Figure 7—figure supplement 3).

These results confirm that *Foxg1-Cre; Cilk1(Ick)* cKO mutants reproduce the phenotypes of *Cilk1(Ick)* KO mutants, including abnormal elongation of cilia, disruption of SHH signaling, shortening of the cochlea, and abnormal apical cochlear specification. These new data have been included in Figure 7—figure supplements 1, 2, and 3 in the revised manuscript.